# Scaling Laws for Forgetting during Finetuning with Pretraining Data Injection

**Louis Bethune** [1] **David Grangier** [1] **Dan Busbridge** [1] **Eleonora Gualdoni** [1] **Marco Cuturi** [1] **Pierre Ablin** [1]

## Abstract

A widespread strategy to obtain a language model that performs well on a target domain is to finetune a pretrained model to perform unsupervised next-token prediction on data from that target domain. Finetuning presents two challenges: *(i)* if the amount of target data is limited, as in most practical applications, the model will quickly overfit, and *(ii)* the model will drift away from the original model, forgetting the pretraining data and the generic knowledge that comes with it. Our goal is to derive scaling laws that quantify these two phenomena for various target domains, amounts of available target data, and model scales. We measure the efficiency of injecting pretraining data into the finetuning data mixture to avoid forgetting and mitigate overfitting. A key practical takeaway from our study is that injecting as little as $1\%$ of pretraining data in the finetuning data mixture prevents the model from forgetting the pretraining set.

## 1. Introduction

Large Language Models (LLMs) are generalist models that are trained on a large and diverse corpus of data, called the pretraining set. The diversity of the pretraining set is widely credited with giving LLMs general knowledge and versatile applicability (Yu et al., 2024). However, not every user requires a gigantic generalist LLM to answer simple tasks, and one might instead favor more specialized models that are able to work better on a specific subset of tasks. For such scenarios, finetuning is the go-to method to obtain a custom LLM that performs well on a comparatively smaller target domain. More precisely, we refer to finetuning in this work as the optimization of the next-token-prediction loss on target domain data, where the starting point of optimization is a pretrained reference model. This definition of finetuning

encompasses the popular supervised finetuning framework (Ouyang et al., 2022), where the target domain data consists of high-quality examples that the LLM should imitate.

**Challenges in Finetuning** Finetuning runs into two intertwined issues. Because the target domain data is often available in limited quantity, notably when compared to the large capacity of an LLM, the model will overfit the target domain during optimization of the training loss. Furthermore, the parameters can move far away from the base models' parameters, which may lead to forgetting: the models' performance on generic tasks decreases during finetuning.

**Pretraining data injection** A customary approach to mitigate forgetting during finetuning consists of injecting some pretraining data into the finetuning data mixture (Liu et al., 2022; Kang et al., 2024; Ibrahim et al., 2024). The idea is that this data serves as a regularizer that helps the model stay good on generic tasks. However, it is unclear how to pick the right amount of pretraining data to inject and what precise impact it has on both forgetting and overfitting.

**Our findings** We consider the target loss, i.e., the loss on a validation version of the target dataset, as a measure of *performance* (relative to the target domain) and the pretraining loss as a measure of *forgetting*. The main takeaway of our work is that for each target domain considered in this paper, the **pretraining loss after finetuning can be predicted accurately** from *(i)* the model scale, *(ii)* the amount of target data available and *(iii)* the fraction of pretraining data injected in the finetuning data mixture. We demonstrate that the following scaling law well predicts the pretraining loss *after* finetuning

$$\mathcal{L}_{\text{pt}} = \underbrace{\mathcal{L}_{\text{pt}}^0}_{\text{Pretraining Loss Before Finetuning}} + A \frac{\overbrace{D_{\text{ft}}}^{\text{Unique Finetune Tokens}}{}^{\beta}}{((1 + \underbrace{B}_{\text{Parameter Relative Efficiency}} \underbrace{p}_{\text{Pretraining Mixture}})N)^{\alpha}}, \quad (1)$$

where $A$, $B$, $\alpha$ and $\beta$ are domain-dependent constants, $N$ is the model size, $D_{\text{ft}}$ is the number of available finetuning tokens, and $p \in [0, 1]$ is the proportion of pretraining data injected in the finetuning data mixture. In particular, we report that, as a rule of thumb, as little as $p = 1\%$ of pretraining data injection is enough to mitigate forgetting.

We also study the impact of these three quantities on the

---

*Equal contribution [1]Apple. Correspondence to: Louis Bethune <l_bethune@apple.com>, Pierre Ablin <p_ablin@apple.com>.

*Proceedings of the 42nd International Conference on Machine Learning*, Vancouver, Canada. PMLR 267, 2025. Copyright 2025 by the author(s).

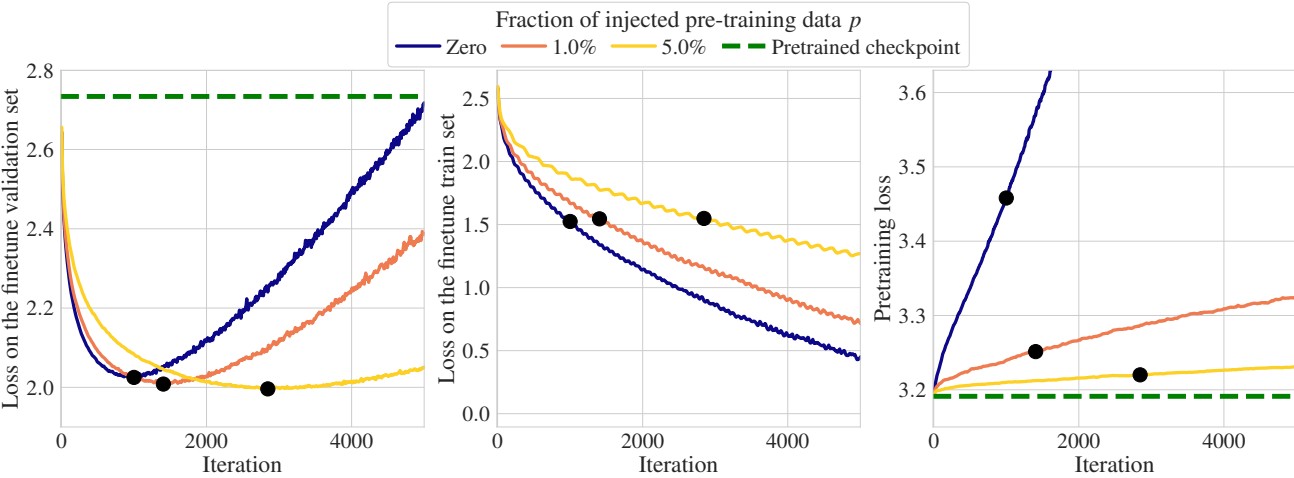

*Figure 1.* **As little as $p = 1\%$ of pretraining data injection shields the model from forgetting on the pretrain dataset**. The finetuning validation follows a conventional U-curve. In this paper, we always consider the models obtained at the bottom of the U-curve, that is, models with the best validation loss on the finetuning set, indicated here by a black dot. Github dataset with small model. The minimum validation loss is barely impacted by the amount of injected pretraining data $p$, and it takes more iterations to reach the minimum as $p$ increases. The loss on the training finetuning set converges to zero as training progresses since the network memorizes the dataset. The pretraining loss increases monotonically during finetuning. **Injecting pretraining data has a regularizing effect that reduces overfitting and forgetting**.

target loss, and report that the data mixture coefficient $p$ has little effect on it. In the case where $p$ is small, we recover the multiplicative scaling laws of Zhang et al. (2024):

$$\mathcal{L}_{\text{ft}} = A \frac{1}{N^{\alpha}} \frac{1}{D_{\text{ft}}^{\beta}} + E, \qquad (2)$$

Parameter Count↑     ↑Unique Finetune Tokens

**Paper organization** In section 2, we describe the models used in this study, the different phases of model training and the pretraining data injection method. In section 3, we describe the models and datasets used in this study, as well as the experimental setup. Finally, section 4 describes our findings and the estimated scaling laws.

### 1.1. Related works

**Auto-regressive pretraining and fine tuning.** Auto-regressive pretraining is a paradigm in which a generative model is trained on sequences to predict the next token given a sequence prefix (Sutskever, 2014). This is typically formulated as a classification task over a given vocabulary built from a *tokenizer*. Finetuning consists of pursuing the auto-regressive pretraining on a specific *finetuning dataset*, orders of magnitude smaller than the pretraining dataset. This fine-tuning dataset typically encompasses data from a single domain, exhibiting a significant distribution shift compared to the base dataset. Transferring the model knowledge to the new distributions can lead to significant capability loss on the previous dataset, a phenomenon known as *forgetting* (Luo et al., 2023; Kalajdzievski, 2024).

**Continual pretraining.** In the context of *continual pre-training*, it has been observed by Ibrahim et al. (2024) that mixing a small percentage of pretraining data mitigates forgetting (see Hiratani, 2024, for an analytical framework to explore how task similarity can affect knowledge transfer). Unlike finetuning, continual pre-training assumes an infinite data stream for the new task, which plays nicely with learning rate schedulers. The scarcity of data in fine-tuning forces repetitions of multiple epochs, which can cause overfitting. Therefore, practitioners typically perform early stopping with a small constant learning rate. Our study focuses on this last scenario.

Alternative approaches in the context of parameter-efficient finetuning (PEFT) have tried to address these challenges by introducing lightweight modules, such as *adapters*, that are updated with knowledge about the new task (Houlsby et al., 2019; He et al., 2022). In this context, Low-Rank Adaptation methods, or LoRA (Hu et al., 2021; Zhang et al., 2024), inject trainable low-rank matrices into the model layers, by adding them to the untouched original model weights, thus reducing the parameter overhead needed to train for new downstream tasks (see Zhu et al., 2024, for a discussion on the different roles of LoRA decomposition matrices). Zhang et al. (2024) question the use of PEFT methods for fine-tuning, reporting that it usually underperforms in terms of fine-tuning loss compared to full-parameter tuning. This is why in this work, we focus on full parameter tuning, and defer the study of PEFT on forgetting for future work.

**Neural scaling laws** were first introduced by the semi-

nal work of Hestness et al. (2017) to predict the final loss achieved by a model, as a function of its number of parameters $N$ and the amount of data $D$ seen. Later, this empirical study was scaled to GPT-2 scale models trained on billions of tokens by Kaplan et al. (2020). In their most simple forms, these "additive" laws are typically written as:

$$L = E + \frac{A}{N^\alpha} + \frac{B}{D^\beta}, \qquad (3)$$

where $E, A, B, \alpha, \beta$ are parameters estimated from measurements. Scaling laws allow us to find the best performance at *IsoFLOPS*, i.e., the lowest loss achievable when the number of FLOPS is fixed. For transformer-like architectures, training FLOPS is approximated as $6ND$ while inference cost is estimated as $2ND$. When $\alpha \approx \beta$, as found in the setup of Hoffmann et al. (2022b), every parameter is worth a constant amount of tokens. In their study, it is a factor $\times 20$ but can go as high as $\times 192$ for training procedures tailored for small models (Hu et al., 2024). This exact value varies from study to study and typically depends on optimizer hyperparameters (Porian et al., 2024; Besiroglu et al., 2024) or the quality of data (DeepSeek-AI, 2024). When accounting for the cost of inference (Sardana et al., 2024), smaller models trained for longer should be preferred over bigger ones. Other scaling laws can be baked in, accounting for mixing different modalities (Aghajanyan et al., 2023), encompassing learning rate scheduling (Tissue et al., 2024) or taking inspiration from statistical physics (An et al., 2024).

**Scaling laws for finetuning.** Conventional training laws focus on datasets too big for overfitting, sometimes even too big to repeat any data. In this context, validation and training loss tend to be equal. This setting cannot be applied in the context of finetuning, where the smaller datasets are subject to *overfitting*, and where the discrepancy between tasks might induce a significant performance gap. This setup of repeating training data has been first studied by Hernandez et al. (2021) and extended by Muennighoff et al. (2023): they apply an exponential "decay" factor $1 - \exp(-R_D/R_D^*)$ to tokens $D$ to account for the lack of information provided by further repetitions $R_D$ over multiple epochs. In Zhang et al. (2024), scaling laws are derived for the finetuning loss with full parameter, LORA adaptation, and prompt finetuning. Other studies focus on the diversity of domains or different measures than the loss (Barnett, 2024; Isik et al., 2024). Close to our work, Kalajdzievski (2024) characterizes forgetting during finetuning. The key differences with our work are that Kalajdzievski (2024) uses an instructed pretrained LLama2 model. In contrast, we use several model scales to characterize the behavior of finetuning at different scales. Also, they measure the forgetting between a pair of datasets, finetuning on one domain and computing the loss on another domain. In contrast, we measure forgetting by looking at the pretraining loss and considering 6 different domains. They also do not mention

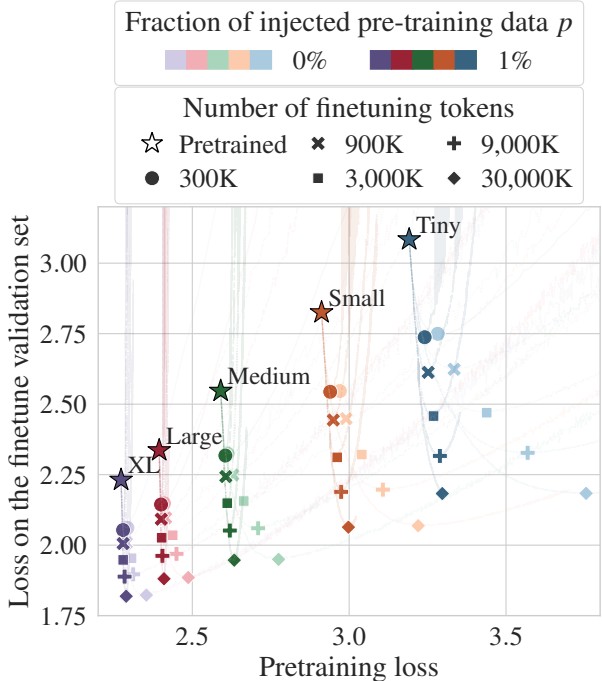

*Figure 2.* **Generalization-memorization tradeoff.** Arxiv domain. Each point corresponds to the bottom of the U-curve for a model trained on datasets of sizes 300K, 900K, 3,000K, 9,000K and 30,000K tokens with mixture parameter $p = 1\%$. **Forgetting is more severe when the model is small and when the finetuning dataset is big**. As shown in Equation 6, this can be attributed to the lack of capacity of the model. More parameters are assigned to training set memorization, and fewer parameters are assigned to the pretraining set performance.

pretraining data injection, which is a key contribution to this work. Finally, they consider parameter-efficient finetuning, while we consider full-parameter finetuning, which is more costly but also more efficient (Zhang et al., 2024).

## 2. Methods

We describe the training phases a model undergoes in our study: pretraining and finetuning. We also describe pretraining data injection for finetuning.

### 2.1. Pretraining phase

The model is trained on a significant amount of tokens from a pretraining dataset $\mathcal{D}_{\text{pretrain}}$ using the next-token prediction loss, until satisfying performance is achieved. The model minimizes the pretraining loss

$$\mathcal{L}_{\text{pt}} = \mathbb{E}_{x \sim \mathcal{D}_{\text{pretrain}}}[\ell(x, \theta)]. \qquad (4)$$

The model is typically pretrained using Adam and decaying learning rate, for example, following a cosine scheduling (Loshchilov & Hutter, 2017) or a square root

one (Hägele et al., 2024). This pretraining loss is low when the model is able to correctly do next-token-prediction on the pretraining set, which, if it is diverse enough, means that the model has general knowledge. In the rest of the paper, we use this pretraining loss as a proxy of general knowledge of a model, and in particular, we measure the forgetting of a model through the increase of pretraining loss.

## 2.2. Finetuning phase

In the second phase of training, the model is finetuned on a smaller task-specific dataset $\mathcal{D}_{\text{finetune}}$, by resuming training from the pretrained model. The parameters are updated by minimizing the loss on $\mathcal{D}_{\text{finetune}}$

$$\mathcal{L}_{\text{ft}} = \mathbb{E}_{x \sim \mathcal{D}_{\text{finetune}}}[\ell(x, \theta)], \tag{5}$$

Since the finetuning set is small, the optimization of this loss quickly leads to overfitting, where the loss on a validation finetuning set increases - this yields a U-curve as displayed in Figure 1. Furthermore, since the model is now only trained by seeing data from the finetuning set, it drifts away from the base model and forgets some of the pretraining set. The optimization of the finetuning loss is typically carried out until the validation loss starts increasing, and this checkpoint gives the so-called finetuned model.

## 2.3. Pretraining data injection

A simple method to mitigate overfitting and model forgetting consists of mixing data from the pretraining set and finetuning the set during finetuning.

Formally, give a mixture parameter $p \in [0, 1]$, we define the mixture of domains $\text{mix}(p) = (1 - p)\mathcal{D}_{\text{finetune}} + p\mathcal{D}_{\text{pretrain}}$. In other words, the probability of selecting a datapoint from $\text{mix}(p)$ is

$$P(x|\text{mix}(p)) = (1 - p)P(x|\mathcal{D}_{\text{finetune}}) + pP(x|\mathcal{D}_{\text{pretrain}}).$$

We can sample efficiently from $\text{mix}(p)$ by first sampling a random binary variable $i \in \{0, 1\}$ with probability $p$ that $i = 1$, and then sampling from $\mathcal{D}_{\text{pretrain}}$ if $i = 1$ and from $\mathcal{D}_{\text{finetune}}$ otherwise. We can then perform finetuning of the model by minimizing the loss over $\text{mix}(p)$:

$$\mathcal{L}_{\text{mix}} = \mathbb{E}_{x \sim \text{mix}(p)}[\ell(x, \theta)] = (1 - p)\mathcal{L}_{\text{ft}} + p\mathcal{L}_{\text{pt}}.$$

This formulation makes it clear that pretraining data injection can be interpreted as adding the pretraining loss as a regularizer to the finetuning loss (Hastie et al., 2017). This way, the model still sees some samples from the pretraining set, which acts as a way to mitigate forgetting and overfitting.

# 3. Experiments

## 3.1. Experimental setup

**Models.** We train GPT2 style transformers (Radford et al., 2019) of different scales. Table 1 reports the model architectures used here. The parameter count follows an exponential progression to cover different magnitudes. For every model, we fix the vocabulary size to $32,000$ and the sequence length to $1,024$. We use the SentencePiece tokenizer (Kudo, 2018). Every computation is performed in *bfloat16* precision, except for normalization layers and softmax in self-attention, which are computed in *float32* precision, following standard practices (Rabe & Staats, 2021; Wang et al., 2024). The biggest model fits on a single $A100 - 80$GB GPU without sharding, with parameter replication across GPUs to handle large batch sizes. 1 GPU is used for *Tiny* and *Small*, 4 for *Medium*, 8 for *Large* and *XL*.

**Datasets.** We use RedpajamaV2 (Weber et al., 2024) as the pretraining set. We use several domains from The Pile (Gao et al., 2020) as finetuning sets, covering all 5 categories ("academic", "internet", "prose", "dialogue" and "misc"). We artificially keep a limited number of tokens to imitate data scarcity, following a log scale between 300 K and 30,000 K tokens - which typically represents between 300 and 60,000 pages from a book, respectively. We believe this accurately reflects realistic use cases where finetuning is performed on specific internal documentation on a specific topic. Data-scarcity means that repetitions of training data will be necessary (multiple epochs), which ensures that overfitting can be observed.

**Pretraining.** We pretrain each model on RedpajamaV2 using standard hyperparameters and a total count of 100 tokens per parameter. The learning rate follows a linear warmup for 0.5% of the total iterations and then follows a cosine scheduling until the end of the training, with a terminal value that is one-hundredth of the maximum value. We use AdamW with a weight decay of $0.1$, which yielded better results than without weight decay (Figure 17). A gradient clipping of 5 was found sufficient to stabilize training across all model scales.

**Finetuning.** We then finetune the model on several domains from the Pile, using a varying amount of target data and different fractions of injected pretraining data. We perform finetuning for 12K steps, which is sufficient to observe a U-curve on the validation loss in every configuration tested. The learning rate is equal to $1/30$ times the peak pretraining LR, which was reached at about $90\%$ of the pretraining stage. Empirically, we observe in the ablations of Figure 5 that this rule of thumb is sufficient to ensure both overfitting well within 12K steps and stable training at all model scales and all mixtures $p$. This corresponds to anywhere between 13 and 5200 epochs, depending on model and data scale.

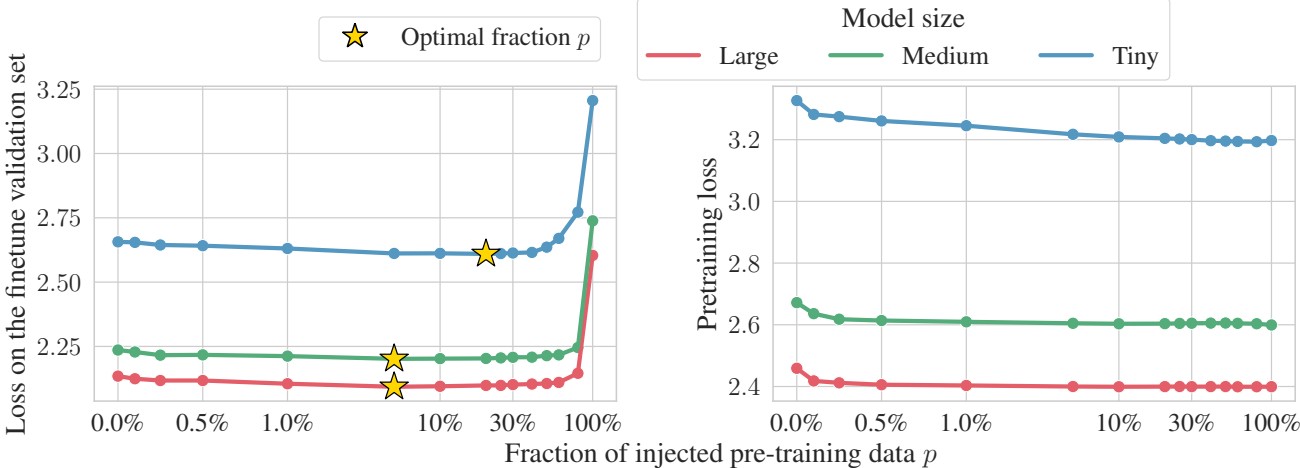

*Figure 3.* **Losses as a function of the fraction of injected pretraining data** $p$ **on Enron emails with 900K finetuning tokens.** Data mixing improves generalization when finetuning data is scarce. The diversity of the pretraining dataset biases learning toward features that exhibit higher generalization. The optimal value of $p$ depends on the domain, the dataset size, and the model size. The finetuning loss as a function of $p$ also follows a U-curve: when $p$ is too small, the model overfits too quickly and does not benefit from the regularizing effect of pretraining data injection. When $p$ is too large, the model does not see enough finetuning data to allocate it enough capacity, there is too much tension with learning from the pretraining set. As expected, increasing $p$ monotonically decreases the pretraining loss.

| Model Size | Parameters (N) | Dim | Heads | Layers | Batch Size | Initial LR | Tokens (D) | FLOPs |
|---|---|---|---|---|---|---|---|---|
| **Tiny** | 41M | 512 | 8 | 8 | 32 | 1e-3 | 5.1B | $1.25 \times 10^{18}$ |
| **Small** | 109M | 768 | 12 | 12 | 32 | 1e-3 | 12.B | $7.84 \times 10^{18}$ |
| **Medium** | 334M | 1024 | 16 | 24 | 64 | 1e-3 | 33B | $6.61 \times 10^{19}$ |
| **Large** | 665M | 1536 | 16 | 24 | 128 | 3e-4 | 66B | $2.63 \times 10^{20}$ |
| **XL** | 1.27B | 2048 | 16 | 24 | 112 | 3e-4 | 100B | $7.62 \times 10^{20}$ |

*Table 1.* **Pretrained model configurations for different sizes of GPT models.** All models use the same tokenizer with a vocabulary size of $32,000$, and the MLP hidden dimension is $4$ times the dimension of the model. The output embedding layer is included in the parameter count, as per Porian et al. (2024). Training FLOPS are reported as $6ND$, following standard practices (Kaplan et al., 2020).

We use Adam without weight decay. The parameter $p$ is chosen from the set $\{0\%, 0.1\%, 0.5\%, 1\%, 5\%\}$.

In total, we, therefore, have 5 model sizes $\times$ 5 number of finetuning tokens $\times$ 5 injection fractions $\times$ 12 domains, which gives 1500 finetuning runs to study.

We report validation losses on the pretraining and target domains, and train losses on the target domains. We use these losses as a proxy for the model's performance on the finetuning domains.

## 4. Results

Unless specified otherwise, we report these values at the checkpoint corresponding to the lower validation domain target loss, i.e., the bottom of the "U-curve".

### 4.1. Mundane observations: overfitting

The loss profile of a typical finetuning run is shown in Figure 1. Unsurprisingly, every validation curve for the finetun-

ing dataset follows a U-shape since the small number of tokens allows for memorization on the train set. The loss over the train set decreases monotonically, with a steeper slope for smaller train sets. The pretraining loss typically follows a forgetting pattern, which is mitigated by taking $p > 0$.

### 4.2. Speed of forgetting

Empirically, we observe in Figure 1 that as little as $p = \mathbf{1\%}$ of pretraining data shields the model from forgetting at no cost for performance on the finetuning dataset. The pretraining data plays an "anchoring" role that helps retain useful features.

On the contrary, in the absence of pretraining data injection, performance can plummet drastically, especially for smaller models. We hypothesize that smaller models must balance their capacity between tasks, whereas bigger ones can "learn a new task" while preserving most existing knowledge.

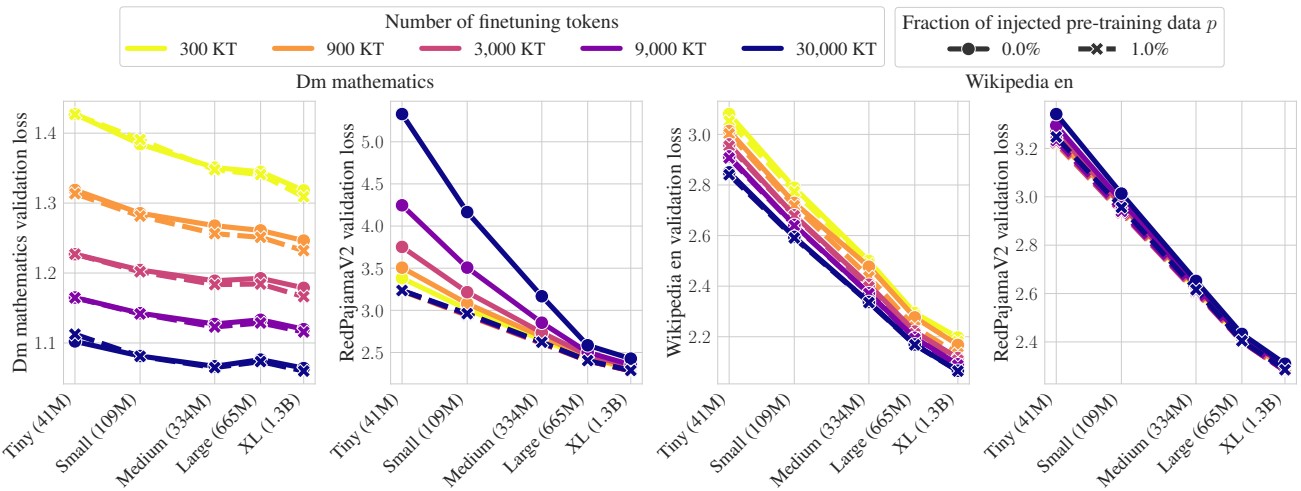

*Figure 4.* **Overfitting and forgetting profiles for two domains.** On one hand, **Dm mathematics (left)** is a dataset that differs a lot from the pretraining set, benefits little from more parameters, and a lot from more data. On the other hand, **Wikipedia En (right)** is more similar to the pretraining set, and more parameters are more beneficial than more training data. Datasets that are far from the pretraining distribution are more prone to forgetting and benefit the most from injecting pre-training data $p > 0$.

| Domain | Finetuning Scaling Laws | | | | | Forgetting Scaling Laws | | | | |
|---|---|---|---|---|---|---|---|---|---|---|
| | $\alpha$ | $\beta$ | A | E | Bootstrapped MRE ($\downarrow$) | $\alpha$ | $\beta$ | A | B | Bootstrapped MRE ($\downarrow$) |
| Arxiv | 0.17 | 0.10 | 95.18 | 1.30 | 0.91% | 0.74 | 0.34 | 526 | 392 | 0.36% |
| Dm mathematics | 0.06 | 0.19 | 16.03 | 0.88 | 0.50% | 0.58 | 0.27 | 202 | 9847 | 0.91% |
| Enron emails | 0.07 | 0.05 | 20.21 | 0.00 | 1.13% | 0.53 | 0.21 | 127 | 1754 | 0.49% |
| Github | 0.14 | 0.12 | 84.55 | 0.79 | 1.40% | 0.76 | 0.43 | 217 | 647 | 0.43% |
| Pg19 | 0.14 | 0.02 | 34.55 | 1.25 | 0.65% | 0.78 | 0.60 | 14 | 259 | 0.39% |
| Wikipedia en | 0.13 | 0.02 | 30.11 | 0.62 | 0.53% | 0.52 | 0.11 | 145 | 829 | 0.21% |
| Euro parl | 0.12 | 0.17 | 160.24 | 1.10 | 1.86% | 0.81 | 0.39 | 2511 | 1107 | 0.79% |
| Free law | 0.19 | 0.05 | 94.06 | 1.11 | 0.91% | 0.75 | 0.45 | 74 | 236 | 0.30% |
| Openwebtext 2 | 0.14 | 0.01 | 31.54 | 0.96 | 0.35% | 0.38 | 0.23 | 2 | 6504 | 0.25% |
| Pubmed abstracts | 0.17 | 0.01 | 46.89 | 0.94 | 0.83% | 0.76 | 0.57 | 8 | 948 | 0.17% |
| Pubmed central | 0.18 | 0.05 | 74.37 | 1.09 | 0.56% | 0.65 | 0.34 | 81 | 574 | 0.26% |
| Stackexchange | 0.16 | 0.08 | 78.23 | 1.27 | 1.03% | 0.62 | 0.34 | 63 | 1179 | 0.27% |

*Table 2.* **Comparison of Scaling Law Coefficients for Finetuning and Forgetting.** The Bootstrapped Estimate of Mean Relative Error $|\hat{y} - y|/y$ across domains for finetuning is **0.89**%, and for forgetting is **0.40**%. These estimates are obtained by resampling each example independently with equal probability to construct a new set of 125 points per domain. The final results are averaged over 128 independent bootstrap repetitions.

### 4.3. Pretraining data injection improves generalization

At $p > 0$, the bottom of the U-curve can reach lower values than $p = 0$, which suggests that pretraining data injection serves are a regularization and helps generalization on the finetuning dataset. This effect can be observed in Figure 3. As a function of $p$, the finetuning validation loss also follows a U-curve - albeit less pronounced. This effect is more striking for small models.

### 4.4. Repeating pretraining tokens during finetuning

In all our experiments, we inject pretraining data that is streamed from the pretraining set without repetition. As reported, injecting only $p = 1\%$ of pretraining data in the training mix is enough to mitigate forgetting significantly. Since $p$ is quite small, it means that we use a small number of pre-training tokens. For instance, in Figure 1, we see that the bottom of the U-curve is reached around 1800 iterations for the run with $p = 1\%$ of injected pre-training data. Since we use a batch-size of 32 and a context length of 1024, it means that, at the bottom of the U-curve, we have only seen

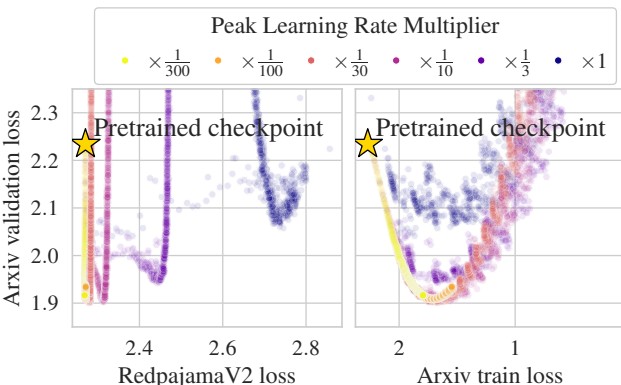

*Figure 5.* **Finetuning learning rate as fraction of peak pretraining learning rate.** Ablation on Arxiv with XL Model (1.3B). The model's learning rate (LR) is reduced by a factor of 100 during pretraining using cosine scheduling. For finetuning, we employ a constant LR, defined as a multiple of the peak LR. Our observations indicate that setting the finetuning LR to $1/30$ times the terminal value strikes an optimal balance between effective adaptation to the finetuning dataset and stability with respect to the pretrained features. Notably, this factor of $1/30\times$ corresponds to the LR reached at 90% of the pretraining phase, near convergence.

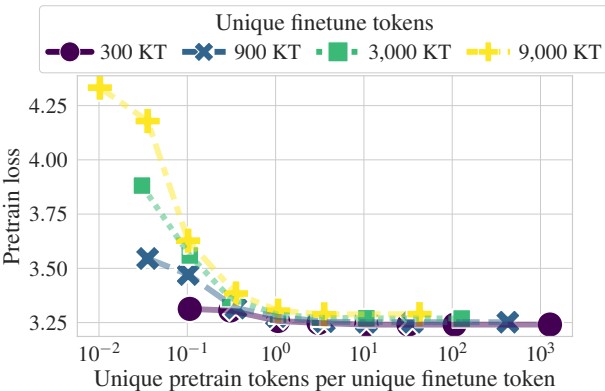

*Figure 6.* **Influence of the number of available pretraining tokens** on the pretraining loss after fine-tuning with $p = 1\%$ of pretraining data injected to the mix. We use the Arxiv domain and a model of size "tiny". With little finetuning data, finetuning is short and hence we can use very few pre-training tokens to reach the optimal forgetting. When there is a lot of finetuning tokens, the optimization takes a long time to reach the bottom of the U-curve. Hence, when the number of pretraining tokens available is also limited, these tokens are repeated many times, which leads to overfitting on the pretraining set as well, and increases the pre-training loss. Remarkably, only 0.3 unique pretraining tokens per unique finetuning token are sufficient to avoid forgetting.

$1\% \times 1800 \times 32 \times 1024 = 600K$ pretraining tokens.

We now turn to a study of how limiting the amount of available *pretraining* tokens that are used for pretraining data injection impacts finetuning and forgetting. To do so, we

consider one domain (arxiv), one model size (tiny), and a fixed fraction of pretraining data injection ($p = 1\%$). We then artificially limit the number of available pretraining tokens in 32KT, 96KT, 320KT, 920KT, and 3,200KT. We report the results in Figure 6. We observe that as the number of available pretraining tokens increases, the pretraining loss after finetuning decreases. However, in this regime, the pretraining dataset itself becomes susceptible to overfitting. This underlines the pivotal role of pretraining dataset diversity in mitigating forgetting.

### 4.5. Scaling laws

We display the loss as a function of model size and dataset size for two domains in Figure 4. We observe a highly predictable behavior that is amenable to the fitting of a scaling law.

We consider different model sizes, which in turn impact the number of pretraining tokens used to train the model, using 100 tokens per parameter. Hence, we do not explore the full pretraining (N, D) space, restricting ourselves to the isocurve $D = 100N$. We have two main reasons for this approach. First, while analyzing the behavior of losses in relation to both N and the total number of pretraining tokens is insightful, using an isocurve aligns with the most common practice, which is to train only one model per model size. Second, pretraining a large grid of models would be computationally expensive, and we prefer to allocate our computing budget toward a more detailed understanding of finetuning mechanisms.

We fit a different scaling law for the finetuning and forgetting loss for every domain. Per domain, we have 5 model sizes × 5 dataset sizes × 5 mixture values = 125 points in total. This is large in front of the number of degrees of freedom of the scaling law (4 or 5 in our case).

**Method.** To fit the coefficients, we follow a conventional approach (Muennighoff et al., 2023), i.e., we rely on the Huber loss and we perform optimization in log space for improved numerical stability. See Appendix C for a detailed discussion.

**Scaling law for finetuning loss** For every domain $\mathcal{D}$ of the Pile, we fit the multiplicative law proposed by Zhang et al. (2024) for finetuning:

$$\mathcal{L}_{\text{ft}} = A\frac{1}{N^\alpha}\frac{1}{D_{\text{ft}}{}^\beta} + E, \qquad (6)$$

which yields a Bootstrapped ($n = 128$) Mean Relative Error (MRE) of **0.89**% across domains. As depicted in Figure 3, the fine-tuning loss is barely impacted by the fraction of injected pre-training data $p$, which is why we chose to treat the loss as a constant of $p$. We report scaling

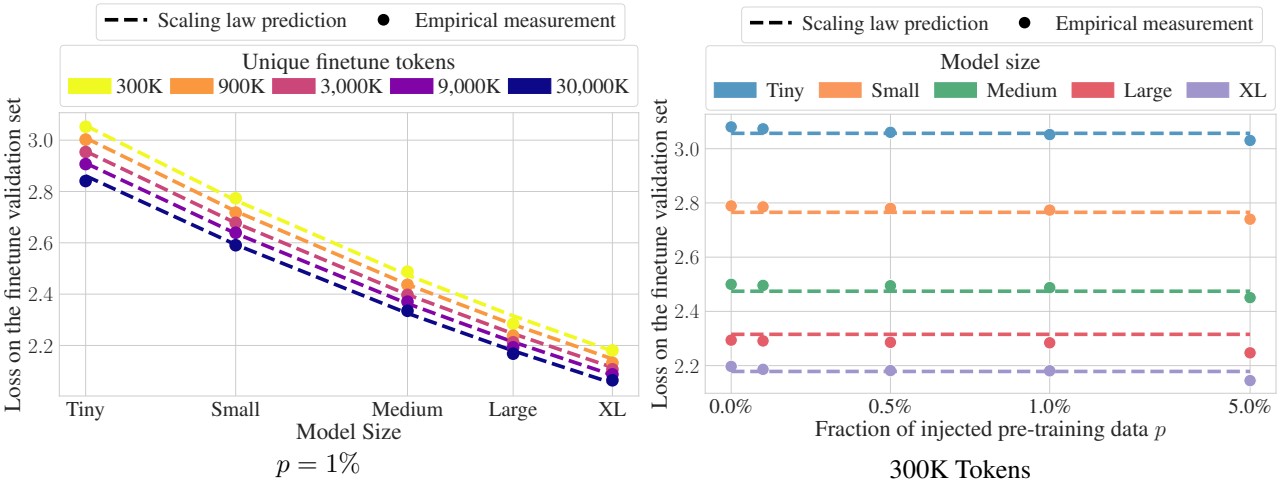

*Figure 7.* **Scaling laws for finetuning loss.** Wikipedia domain. We extend the multiplicative laws of Zhang et al. (2024) to take into account the fraction of injected pretraining data. At first order, this scaling law is independent of the mixture $p$. **Left** Agreement between the observed behavior and the behavior predicted by the scaling law. **Right** Evaluated and predicted loss as a function of $p$.

laws coefficients in Table 2. The results for fixed dataset size or for a fixed mixture $p$ are given in Figure 7. We also tried to fit an additive law of the form $\mathcal{L}_{\text{ft}} = \frac{A}{N^{\alpha}} + \frac{B}{D_{\text{ft}}^{\beta}} + E$, but despite the additional degree of freedom through coefficient $B$, it yields an higher MRE of 1.36%. The superiority of the multiplicative scaling law compared to the additive is consistent with the findings of Zhang et al. (2024).

**Analysis.** The value of $E \approx 0$ for Enron emails stands out in the table. This pattern suggests that memorization of the corpus is possible, firstly thanks to its modest size, and secondly because it has been made available online more than 20 years ago and used in several NLP publications ever since then[1]. Besides that, the value of $\alpha$ oscillates between 0.12 and 0.19, which is consistent with previous studies (Hoffmann et al., 2022a). The value of $\beta$ seems to capture the difficulty of the dataset, with the high-entropy ones like Wikipedia or Openwebtext standing out.

**Scaling law for forgetting** For every domain of the Pile, we want to predict $\mathcal{L}_{\text{pt}}$, the value of the pretraining loss *after* finetuning, as a measure of forgetting. We propose the (modified) multiplicative law:

$$\mathcal{L}_{\text{pt}} = \mathcal{L}_{\text{pt}}^0 + A \frac{D_{\text{ft}}^{\beta}}{((1 + Bp)N)^{\alpha}}, \qquad (7)$$

where $\mathcal{L}_{\text{pt}}^0$ is the pretraining loss *before* fine-tuning, i.e. the pretraining loss of the pretrained model, $D_{\text{ft}}$ is the number of available fine-tuning tokens, $N$ is the model size. The variables of the scaling law are $A, B, \beta$ and $\alpha$, and they are all positive numbers. Note that the value $\mathcal{L}_{\text{pt}}^0$ is exactly what the original scaling law papers try to estimate (Kaplan et al.,

2020; Hoffmann et al., 2022a). We propose a multiplicative law where $D^{\beta}$ is on the *numerator* with $\beta > 0$, since increasing the finetuning dataset size leads to more iterations to reach the bottom of the U-curve, which results in a model that drifts further from the base model and a higher pretraining loss. We use a factor $(1 + Bp)$ in front of the $N$ parameter. It accounts for the fact that a fraction $p$ of the parameters are allocated to the pretraining task rather than the finetuning one, and they are $B$ times more efficient in this context. Typically, we observe $B \gg 1$ since the features of the pretrained model are already aligned with the pretraining task, which translates into more efficient parameter allocation. Coefficients are given in Table 2. Since fine-tuning is performed with a learning rate slightly higher than the terminal LR of pretraining, the value of $\mathcal{L}_{\text{pt}}^0$ corresponds to the *rewarmed* pretrained model, i.e, the model pretrained with the LR used for fine-tuning, and *not* the terminal LR of the cosine scheduling. Differences are highlighted in Table 3. The average Bootstrapped MRE across domains is **0.40%**.

**Other laws.** An additive law to predict the difference $\mathcal{L}_{\text{pt}} - \mathcal{L}_{\text{pt}}^0$ has an error greater than 0.82% despite having one more degree of freedom. We also note that it is critical to add the base pretraining loss $\mathcal{L}_{pt}^0$ to the scaling law, as fitting a scaling law of the form $\mathcal{L}_{pt} = A \frac{D_{\text{ft}}^{\beta}}{((1+Bp)N)^{\alpha}} + E$ leads to a MRE of 1.05%. Finally, we observe that the "cost of rewarming" the model must be accounted for when computing $\mathcal{L}_{\text{pt}}^0$, as observed in Table 3. Otherwise, we find that an additive corrective factor of $E \approx 0.05$ is necessary to account for forgetting induced by optimization, which yields a higher MRE of 0.49%. Indeed, even in the setting $p = 1$, further pretraining with a constant LR (which is typically higher than the terminal LR as shown in Figure 5)

---

[1]See http://www.enron-mail.com for example.

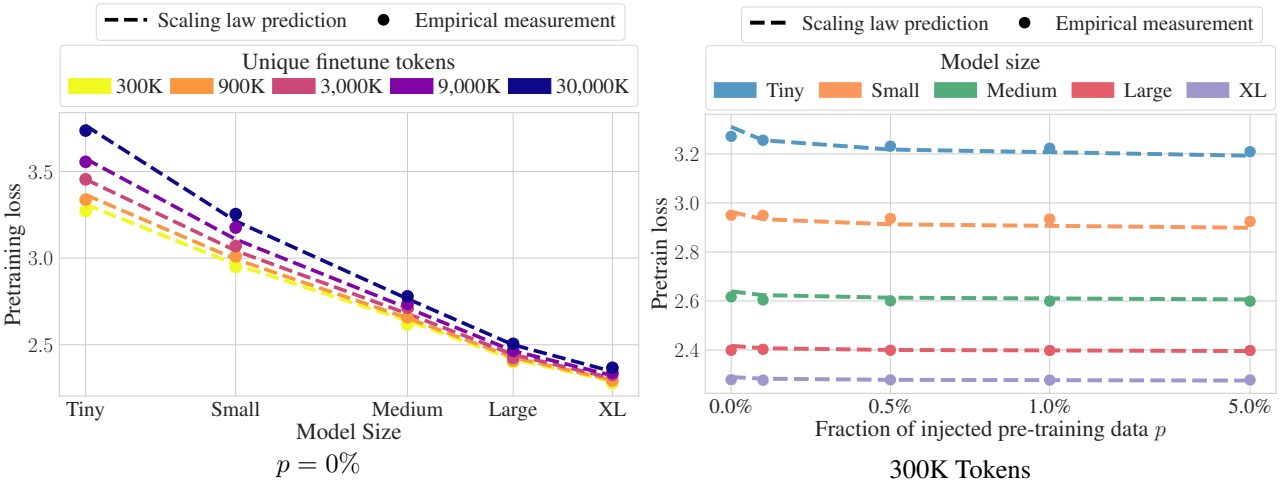

*Figure 8.* **Scaling laws for forgetting.** Github domain. We propose to model the increase in pretraining loss as a multiplicative scaling law (Equation 1), that takes into account model size, number of finetuning tokens available, and fraction of pretraining data injected in the data mixture $p$. **Left** Agreement between evaluated and predicted loss on one domain. **Right** Evaluated and predicted loss as a function of $p$.

| Model Size | Pretraining loss $\mathcal{L}_{\mathrm{pt}}^0$ with | |
| | pretrained model $1/100$ peak LR | rewarmed model $1/30$ peak LR |
|---|---|---|
| **Tiny** | 3.13 | 3.19 |
| **Small** | 2.84 | 2.92 |
| **Medium** | 2.55 | 2.60 |
| **Large** | 2.34 | 2.39 |
| **XL** | 2.22 | 2.27 |
| Forgetting MRE ($\downarrow$) | 0.49% | **0.40**% |

*Table 3.* **Effect of rewarming the model on the pretraining loss.** Finetuning is performed with a learning rate (LR) which is $1/30$ of the peak LR, slightly higher than the terminal LR of the pretraining stage. This "rewarms" the model, incurring a loss increase that we measure and must be accounted for.

cancels some of the benefits of the cosine cooldown used during pretraining.

**Analysis.** The value of $B$ indicates how much pretraining data injection helps mitigate forgetting. For Dm mathematics, which was observed in Figure 4 to be highly subject to forgetting, $B \approx 10^4$. On the contrary, for Wikipedia, on which pre-training data injection was less useful, we only have $B \approx 829$.

An interesting consequence of our functional form is that forgetting is primarily attributed to network capacity. This is confirmed by the fact that smaller models suffer the most: they lose up to 95% (!) of the pretraining progress when forgetting (i.e, the pretraining validation loss reverts to a point reached at 5% of pretraining), while bigger models only lose 20% of the progress. However, bigger models

require more compute, so forgetting is more expensive for them. Results are summarized in Figure 14.

**Extrapolation.** Scaling laws are mainly useful when they enable performance predictions based on small-scale experiments, to assess if bigger models and datasets are worth the cost. To test this, we design a synthetic experiment where scaling law coefficients are fitted using only a subset of model sizes and fine-tuning dataset sizes. The results, presented in Table 4, demonstrate that models trained on no more than *Medium* (334M parameters) and 3,000K tokens can accurately predict the performance of *Large* and *XL* models trained on 9,000K and 30,000K tokens. Averaged over all domains, the prediction error remains within 2.01% on the finetuning set and 0.83% on the pretraining set. On 4 GPUs, the *Medium* model fine-tuned on 3,000K tokens reaches the bottom of the U-curve in under 30 minutes, whereas the *XL* model requires up to 7 hours on 8 GPUs. The cost of acquiring additional data to scale from 3,000K to 30,000K tokens varies depending on the context, but can be significant in certain environments. In this context, scaling laws can lead to substantial computational savings.

## Conclusion

We showed that one can accurately predict the finetuning performance and the forgetting of the pretraining set of large language models, as a function of the model size, the number of available tokens, and the fraction of pretraining data injected into the data mixture. This behavior is very consistent across datasets and can be explained by simple scaling laws. A key takeaway of our work is that injecting even 1% of training data during fine-tuning helps mitigate pretraining set forgetting.

## Acknowledgments

The authors heavily relied on a codebase that Awni Hannun kickstarted. The authors thank Federico Danielli for his careful proof-reading. Finally, the authors warmly thank Alaaeldin El Nouby for his judicious suggestions in the design and organization of the figures and tables.

## Impact Statement

This paper presents work whose goal is to advance the field of Machine Learning. There are many potential societal consequences of our work, none of which we feel must be specifically highlighted here.

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

# A. Ablations

**Learning rate.** The learning rate (Figure 5) is a critical parameter. If it is too low, we might fail to reach the bottom of the U-curve within a reasonable time. If it is too high, it causes the model to diverge away from the pretraining validation loss. Warmup and cosine scheduling cause modelization issues in this context. In particular, they require to know in advance the number of steps necessary for convergence, which is tricky when targeting the bottom of a U-curve. Therefore, we focus on a constant learning rate.

**Anchored AdamW.** We propose another approach to reduce forgetting, coined *Anchored AdamW*, which performs the same updates as AdamW with the difference that the regularization ties the parameters $\theta_t$ to the pre-trained model $\theta_0$, instead of $\mathbf{0}$.

$$\text{regularization} = \frac{\lambda}{2}\|\theta_t - \theta_0\|_2^2. \tag{8}$$

This contrasts with the default weight decay which is simply $\frac{\lambda}{2}\|\theta_t\|_2^2$. The pre-trained parameters $\theta_0$ hence play the role of anchoring. The difference between Adam, AdamW, and Anchored Adamw is illustrated in Figure 9. The results on the Github domain are given in Figure 10. In this scenario, we see that adding only $p = 1\%$ of pre-training data outperforms fine-tuning with weight decay by a large margin. We see that the higher performance of finetuning with pre-training data can not be attributed to closeness with $\theta_0$ only. Anchored AdamW fails to prevent forgetting on RedPajama, but at the same time, the regularization is strong enough to hurt performance on Github noticeably. This suggests that having more data diversity is more powerful than simple parameter-space regularization.

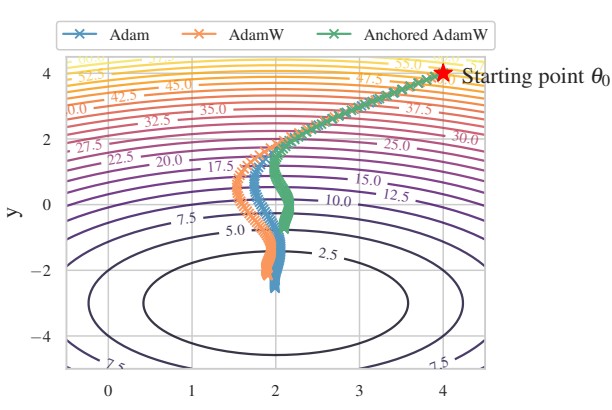
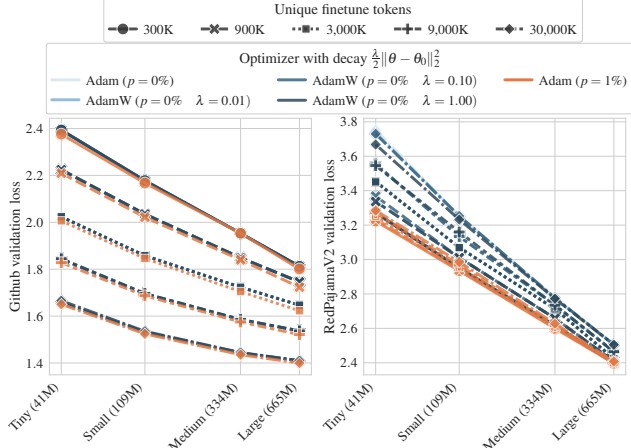

*Figure 9.* Comparison of Adam, AdamW, and Anchored AdamW on the objective function $f(x,y) = (x-2)^2 + (y+3)^2$ with the starting point $\theta_0 = (4,4)$. 100 steps, learning rate 0.1.

*Figure 10.* Comparison of Adam and Anchored AdamW on Github domain. Mixing $p = 1\%$ of pre-training data with Adam optimizers clearly outperforms Anchored AdamW, both on pre-trained data and fine-tuning data.

**Forgetting law.** Other laws can be tested, such as $A\frac{D^\beta(1-p)^\kappa}{N^\alpha} + E$ which has the appealing property of going to zero when $p = 1$. However, the MRE reaches $0.67\%$ for this law, despite having the same degrees of freedom with the new parameter $\kappa$. Purely additive laws had an MRE superior to 1%.

**Additional isocurves.** Most experiments of the paper rely on the isocurve $D = 100N$ for the checkpoint of the pre-trained model. We also re-train full models from scratch with cosine decay for the isocurve $D = 10N$, and evaluate them on the Freelaw domain. Results are given in Figure 11 and demonstrate that our scaling laws still hold for these checkpoints.

**Extrapolation.** In Table 4 we evaluate the extrapolation capabilities of the scaling law, by fitting only on a subset of all available tokens and models. Overall, the MRE remains below 2% for the law of (Zhang et al., 2024), and even below 1% for the forgetting law we propose. This demonstrates the feasibility of estimating coefficients at a small scale and evaluating their impact at a larger scale.

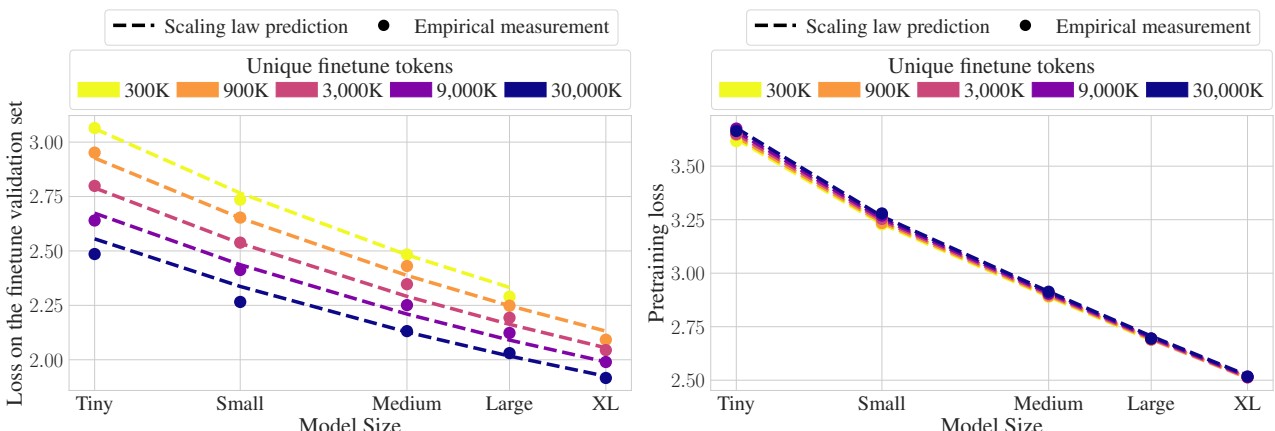

*Figure 11.* **Scaling laws for finetuning and forgetting in models pre-trained at the** $D = 10N$ **isocurve.** Freelaw domain. The law remains robust to the choice of checkpoint, with a bootstrapped MRE of 0.57% for forgetting and 1.14% for finetuning.

| Setup | Predict on | Finetuning MRE | Forgetting MRE |
|-------|-----------|----------------|----------------|
| **A** | XL 30,000K | 1.69% | 0.73% |
| **B** | Large, XL 9,000K, 30,000K | 2.01% | 0.83% |

*Table 4.* **Mean Relative Error across all 12 domains in the extrapolation setting.** In setup **A**, scaling law coefficients are estimated on all model sizes except *XL*, and on all finetuning set sizes except 30,000K tokens. In setup **B**, the largest model used for fitting is *Medium* (334M) and the maximum finetuning dataset size is limited to 3,000K tokens.

**Instruction finetuning (IFT).** Our work focused mainly on the next token prediction task in finetuning, directly on the raw text sequence, but a variant exists: instruction finetuning (Wei et al., 2022; Sanh et al., 2022; Min et al., 2022). In this setup, the "instruction" tokens (sometimes referred to as "prompt" or "input") are masked from the loss, and the model is trained to solve a specific task conditioned by the prompt. Implementation-wise, we leverage that our vocabulary is capped to 32k tokens (ids can be stored on 15 bits), and we store the mask in the (unused) 16th bit of the uint16 types we use to store token ids. At train time, bitwise operations are used to efficiently extract the mask from the ids, and apply the next token prediction loss only on the unmasked tokens. We rely on the OpenHermes dataset (Teknium, 2023), with special "[INST]" and "[/INST]" tokens to delimit the instruction part from the prediction part. Results are given in Figure 12. We fit the scaling law coefficients, and we see that the bootstrap estimate of MAE is 0.59% for finetuning, and 0.29% for forgetting. Coefficients are given in Table below.

| Domain | Finetuning Scaling Law | | | | | Forgetting Scaling Law | | | | |
|--------|-----|-----|-----|-----|-----|-----|-----|-----|-----|-----|
| | $\alpha$ | $\beta$ | A | E | MRE | $\alpha$ | $\beta$ | A | B | MRE |
| OpenHermes | 0.17 | 0.03 | 64.28 | 0.46 | 0.59% | 0.80 | 0.27 | 5513 | 8584 | 0.29% |

# B. Practical consequences on finetuning

**The hidden cost of forgetting.** The Figure 14 shows that small models suffer the most from forgetting, losing up to 95% of the pre-training stage. This makes them unsuitable for sequential adaptation, and they benefit less from pre-training on diverse data when fine-tuning specific data. Overall, this balances some findings of Sardana et al. (2024), which suggested that smaller models should be preferred due to their lower inference cost. While being more expensive, bigger models retain more information from the pre-training stage.

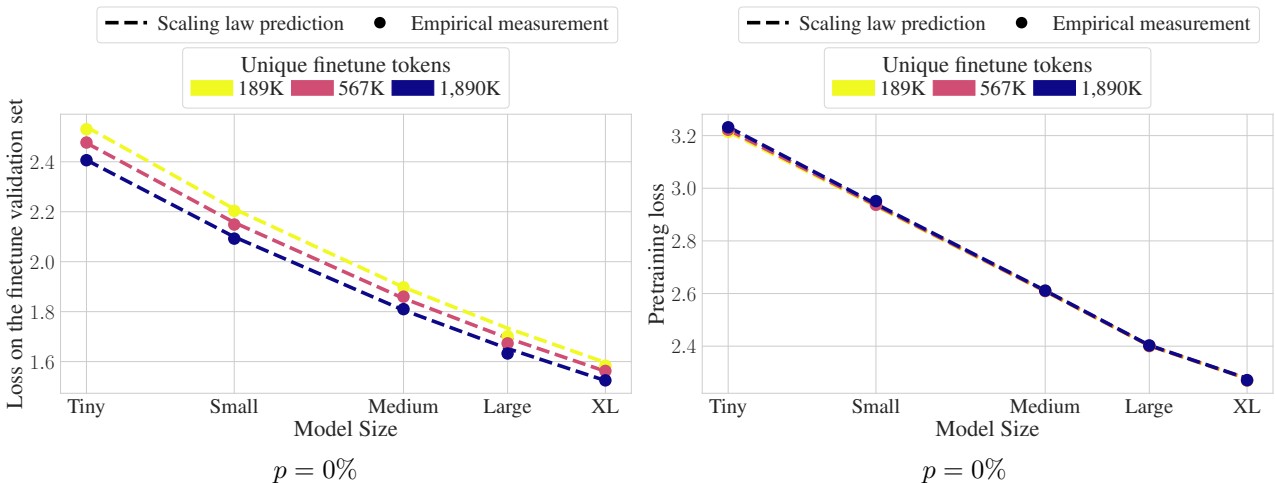

*Figure 12.* **Scaling laws for finetuning and forgetting with Instruction Finetuning (IFT) on OpenHermes dataset.** The dataset is much smaller than the domain of The Pile considered here, so different values of unique tokens are considered. Finally, this dataset is very diverse and challenging by design, which flattens the contribution of the number of tokens, and makes the model size the driving factor behind scaling.

**Downstream task performance.** It has been documented that training loss correlates with downstream task performance (Hoffmann et al., 2022a; Mayilvahanan et al., 2025). Therefore, we evaluate the quality of the 1.3B model, *before* and *after* finetuning, *with* and *without* pre-training data injection, on ARC-easy (Clark et al., 2018) and MMLU tasks (Hendrycks et al., 2021). Results are given in Figure 13. We see that the performance on ARC-easy degrades after fine-tuning, but injection of pre-training data mitigates this phenomenon. On MMLU the trend is less obvious, as the model does not perform better than a random classifier. However, the monotonic improvement across all model scales suggests that dm-mathematics contains an inductive bias that helps on some tasks of MMLU.

Furthermore, for ARC-easy in the 0-shot setting, we compare the performance with pretraining data injection ($p > 0\%$) against the baseline at $p = 0\%$, and we report the difference in accuracy when it's significant at the 99% threshold, based on the z-score. Otherwise, we report the difference as "Not Significant" (n.s). We see that injecting pre-training data yields a consistent improvement across all model scales and data abundance.

| Model size | Number of tokens $D$ | Percentage of pre-training data injection $p$ | | | |
|---|---|---|---|---|---|
| | | 0.1% | 0.5% | 1% | 5% |
| Medium | 300 KT | ns | ns | ns | ns |
| Medium | 900 KT | ns | ns | ns | ns |
| Medium | 3,000 KT | 3.9 | **4.5** | 4.0 | 4.0 |
| Medium | 9,000 KT | 4.1 | 4.1 | 4.0 | ns |
| Medium | 30,000 KT | 4.8 | **6.6** | 4.8 | 5.4 |
| Large | 300 KT | ns | ns | ns | ns |
| Large | 900 KT | ns | ns | ns | ns |
| Large | 3,000 KT | ns | 3.9 | 4.6 | **5.1** |
| Large | 9,000 KT | 3.8 | 5.0 | **5.3** | 4.6 |
| Large | 30,000 KT | 6.9 | 7.7 | 7.2 | **8.4** |
| XL | 300 KT | ns | ns | ns | ns |
| XL | 900 KT | ns | ns | ns | ns |
| XL | 3,000 KT | ns | ns | ns | ns |
| XL | 9,000 KT | ns | ns | 4.0 | ns |
| XL | 30,000 KT | 4.8 | 5.0 | **5.5** | 5.3 |

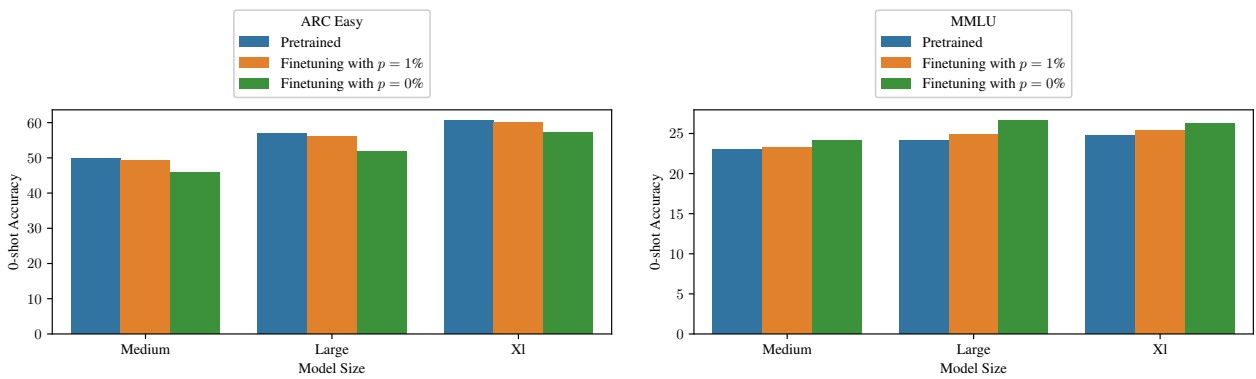

*Figure 13.* **Downstream tasks performance for pretrained checkpoint before finetuning, after finetuning, and with data injection.** All checkpoints have been fine-tuned on dm-mathematics. Forgetting penalizes the model on ARC Easy, but finetuning improves (marginally) the model on MMLU.

## C. Scaling law coefficient estimation

We follow the procedure outlined in (Hoffmann et al., 2022a; Muennighoff et al., 2023; Besiroglu et al., 2024), to identify the coefficients in our scaling laws. Restating the supervised law for convenience

$$L(N, D) = E + \frac{A}{N^\alpha} + \frac{B}{D^\beta}. \tag{9}$$

To aid numerical stability, we write this expression in log space. First note that for $a, b > 0$

$$\log(a + b) = \log\left(\exp\log a + \exp\log b\right) = \mathrm{LSE}(\log a, \log b), \tag{10}$$

where LSE is the log-sum-exp operator. We can now proceed to write the supervised scaling law in log form

$$\log L(N, D; A, B, E, \alpha, \beta) = \log\left[E + \frac{A}{N^\alpha} + \frac{B}{D^\beta}\right] \tag{11}$$

$$= \mathrm{LSE}\left[\log E, \log A - \alpha N, \log B - \beta D\right]. \tag{12}$$

We make no assumptions about the relationships between the values (i.e. *no parameter tying*) and optimize

$$(A^*, B^*, E^*, \alpha^*, \beta^*) = \operatorname*{arg\,min}_{\{A,B,E,\alpha,\beta\}} \sum_i \mathrm{Huber}_\delta\left(\log L(N^{(i)}, D^{(i)}; A, B, E, \alpha, \beta) - L^{(i)}\right) \tag{13}$$

with a Huber $\delta = 10^{-4}$, where $N^{(i)}$, $D^{(i)}$ and $L^{(i)}$ are the model size, number of training tokens and loss achieved by the $i$-th run. We fit on 125 samples over a grid of L-BFGS-B initializations given by: $\log A \in \{0, 3, 6, 9, 12\}$, $\log B \in \{0, 3, 6, 9, 12\}$, $\log E \in \{-2, -1.5, -1, 0, 0.5, 1, 1.5, 2, 2.5, 3\}$, $\alpha \in \{0, 0.5, 1\}$, $\beta \in \{0, 0.5, 1\}$. We use a similar procedure to fit the different forms of the scaling laws.

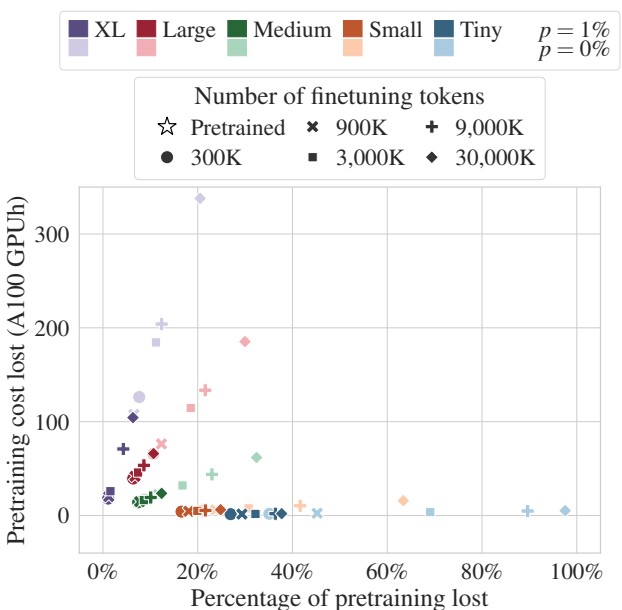

*Figure 14.* **Cost of forgetting.** Points are reported at the bottom of the U-curve for the Arxiv domain. Same setup as Figure 2. Smaller models lose a significant portion of the progress achieved during pretraining—up to 80%—effectively negating much of the initial effort. In contrast, larger models retain more knowledge due to their greater capacity, allowing them to accommodate both tasks simultaneously. However, these models are also the most expensive to train, not only because of their size but also due to the increased number of pretraining tokens required for compute-optimality (Hoffmann et al., 2022a). Consequently, the GPU-hour cost induced by forgetting is substantially higher for larger models.

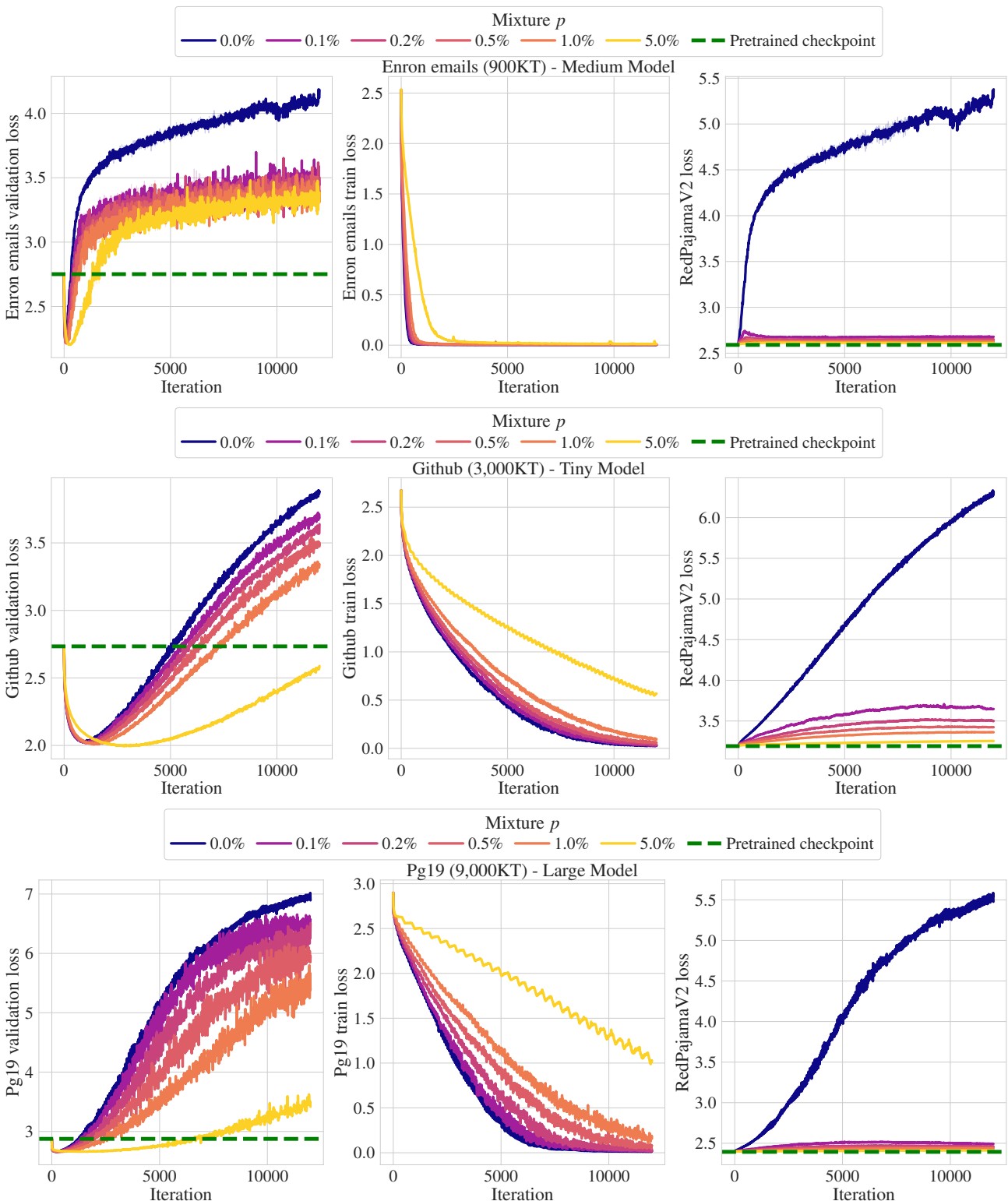

*Figure 15.* **As little as** $p = 1\%$ **of pretraining data shields the model from forgetting on the pretrain dataset**.

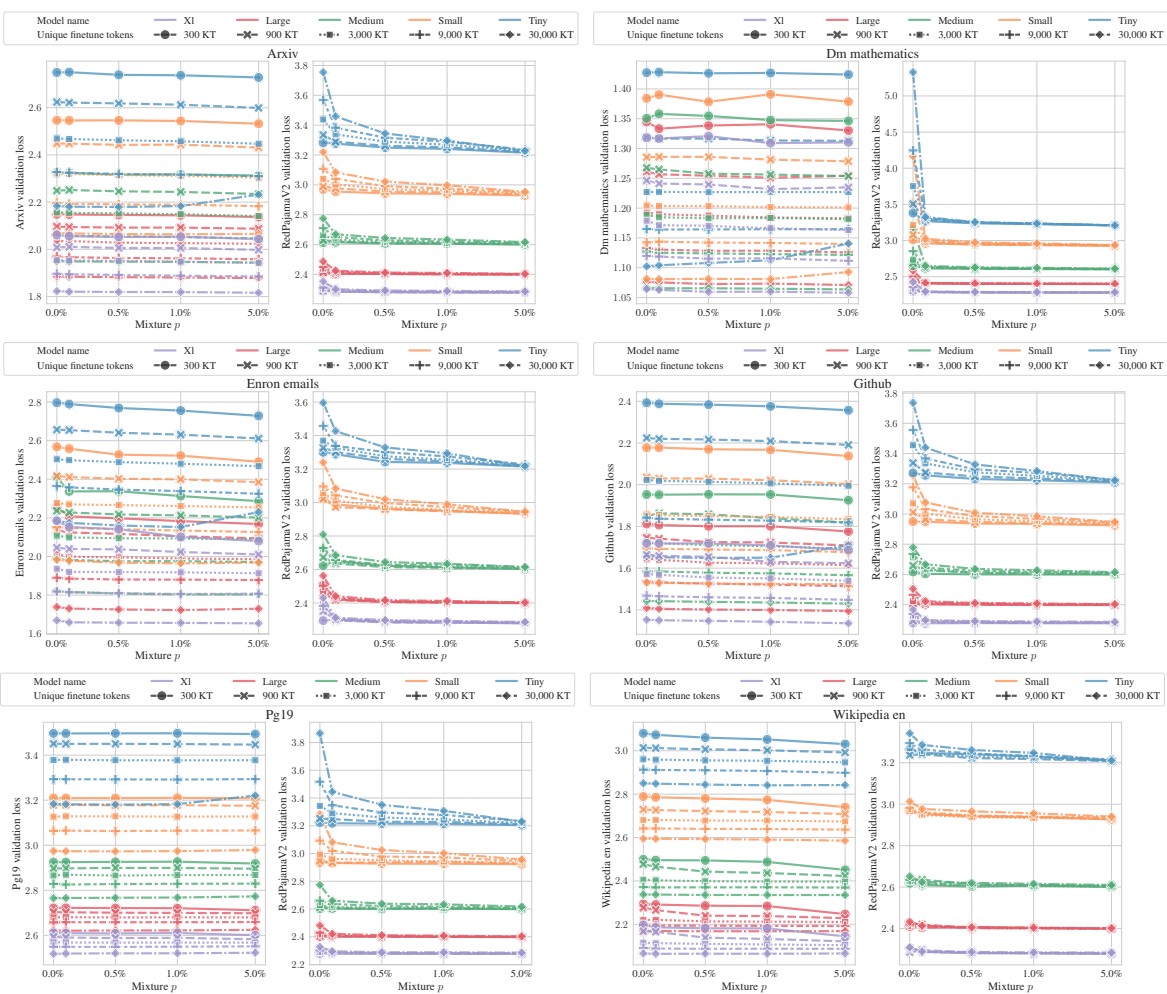

*Figure 16.* **Losses at the bottom of the U-curve** for 6 domains of The Pile with 5 models and 5 dataset sizes, for all values of data mixture $\gamma$.

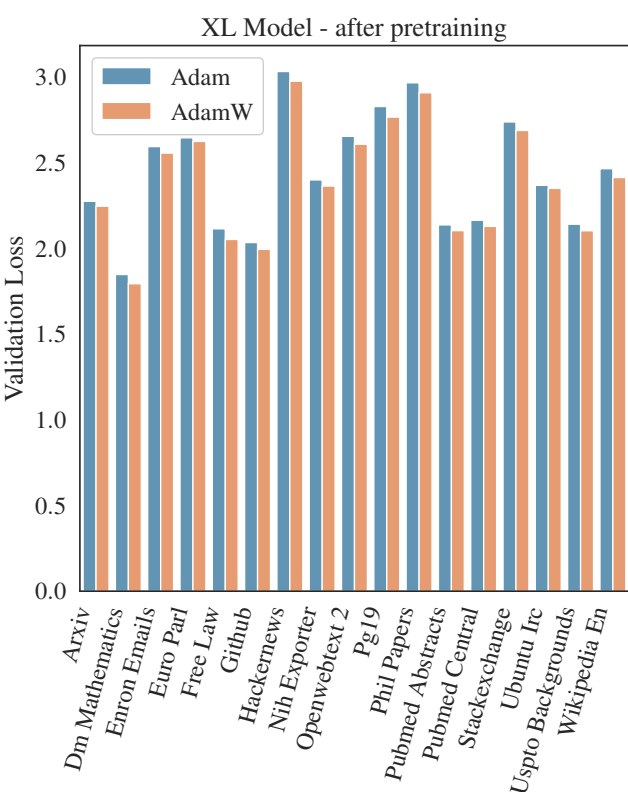

*Figure 17.* **Ablation.** Weight decay during pretraining improves checkpoint quality. Note that default implementations of Pytorch and Optax do not decouple the weight decay from the learning rate (per this Github issue: `https://github.com/google-deepmind/optax/issues/292`), unlike the seminal paper suggested (Loshchilov & Hutter, 2019), which implies that weight decay impact diminishes during learning rate decay.

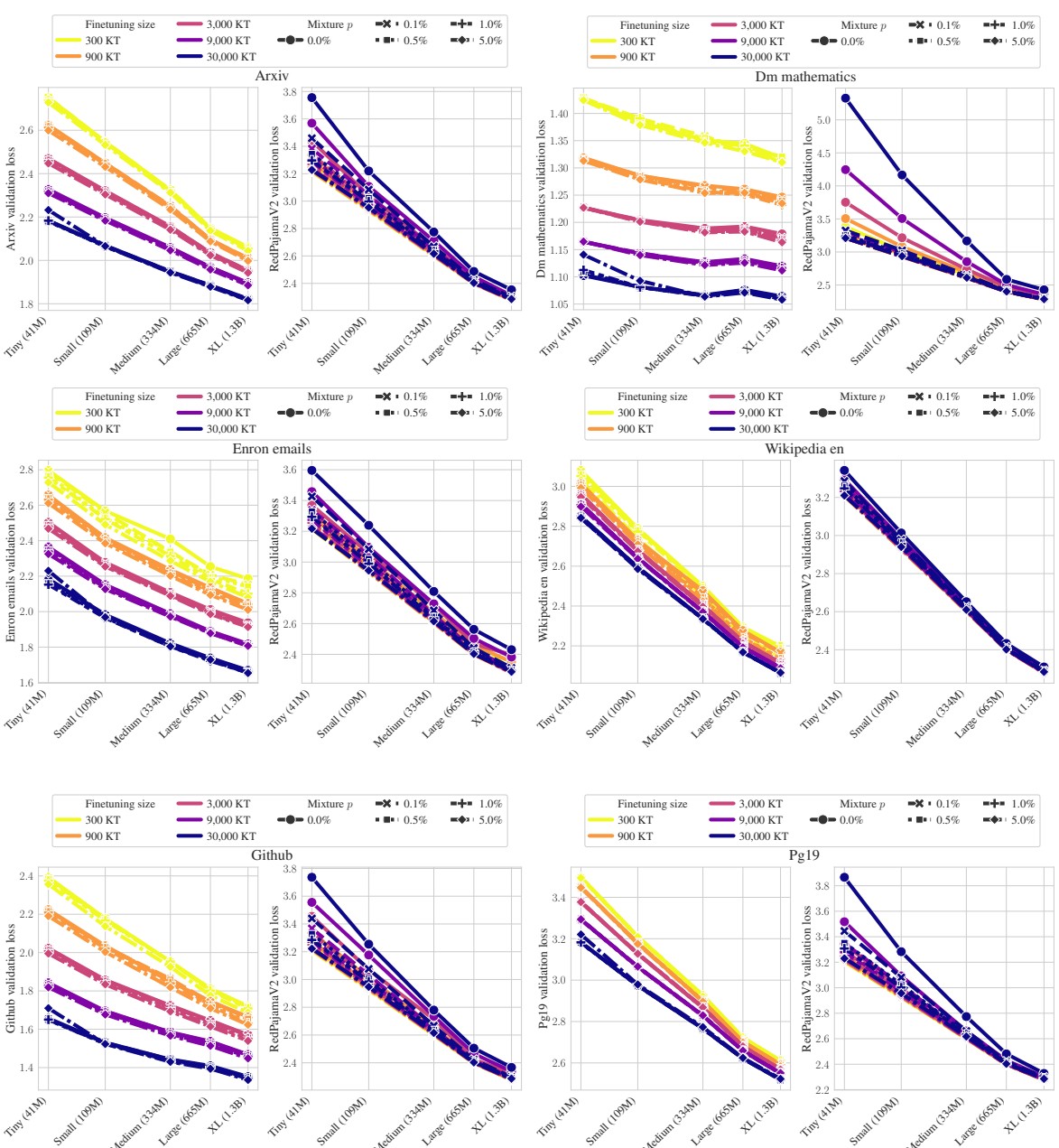

*Figure 18.* **Losses as a function of model size, dataset size, and data-mixing.**

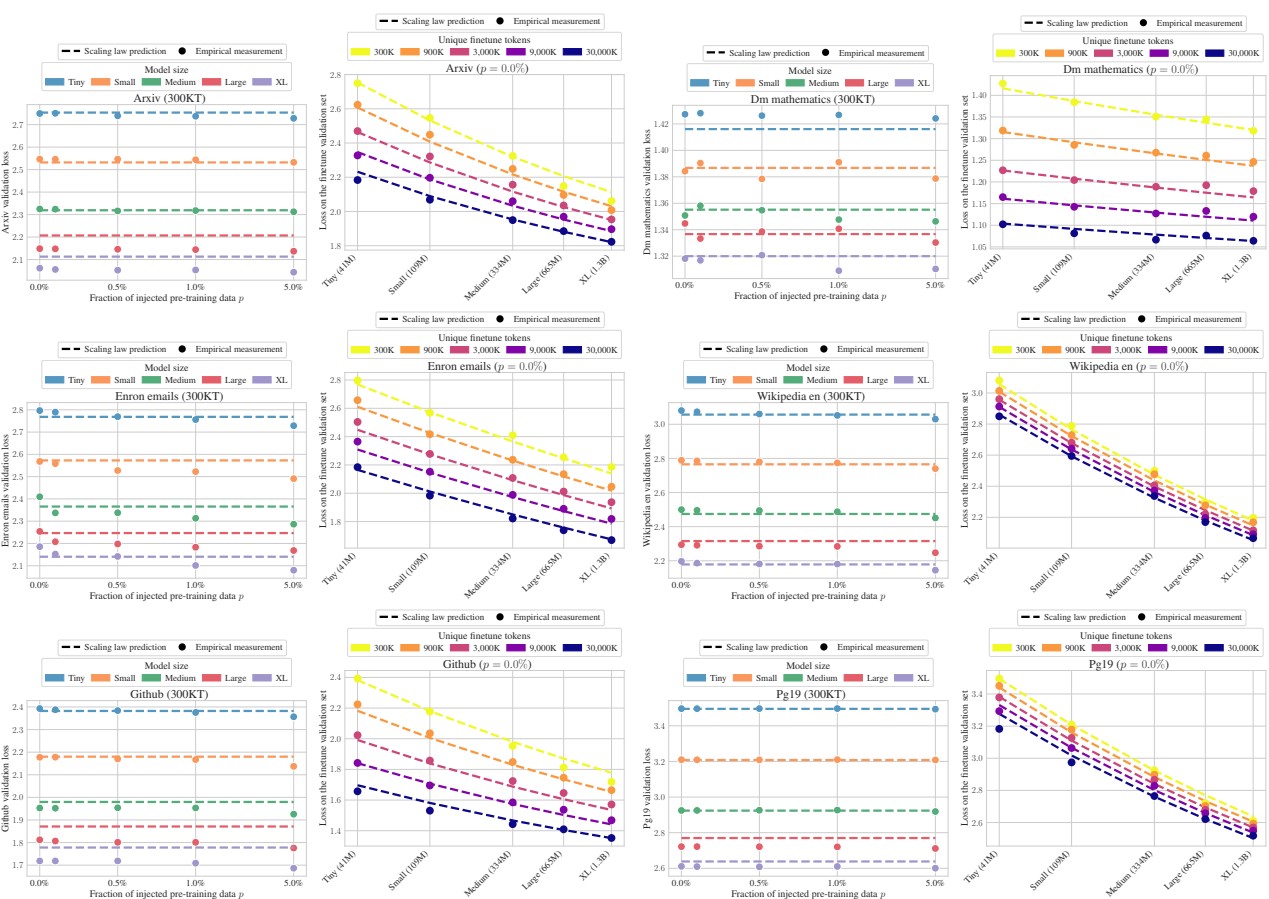

Figure 19. **Example of finetuning scaling laws for several domains.**

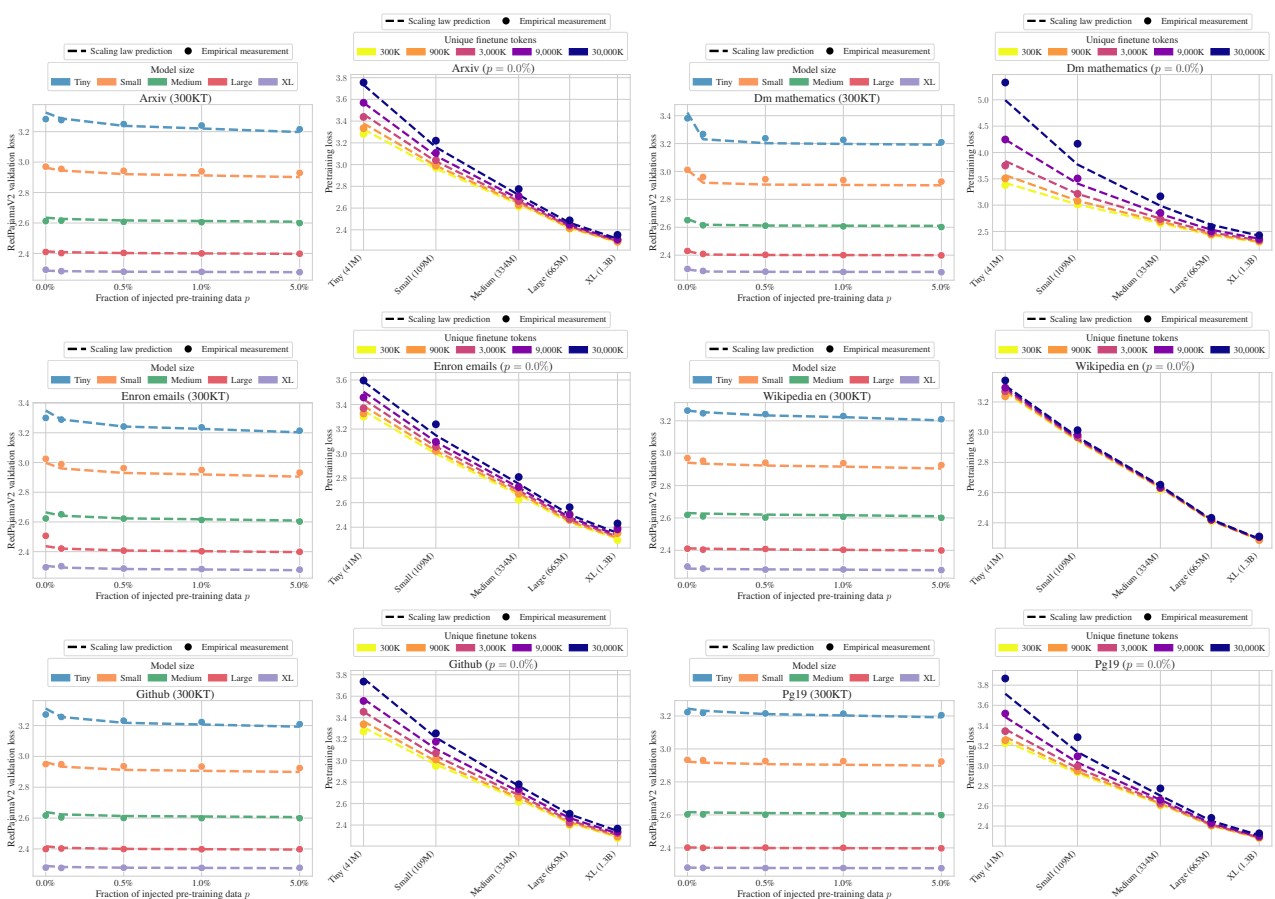

*Figure 20.* **Example of forgetting scaling laws for several domains.**

