# OpenReview forum: "Scaling Laws for Forgetting during Finetuning with Pretraining Data Injection"
_ICML.cc/2025/Conference — ICML 2025 poster_

### Official Review · Reviewer_R4eh · 2025-03-11

**Overall Recommendation:** 4

**Summary:**

This paper presents a study of scaling laws for fine-tuning, in the particular case where replay data (in the form of pretraining data) is available. The paper models the forgetting loss as a function of the replay data, fine-tuning data, and number of parameters.
It also extends the scaling law of existing work on modeling the fine-tuning loss, showing that the amount of injected replay data barely impacts the final validation loss on the fine-tuning data. They also show that as low as $p$=1% of the pre training data is sufficient to not lose performance on pretraining data, and that the scaling coefficient $p$ depends on the nature of the fine-tuning domain. The experiments are comprehensive and validate the hypotheses.

## Update after rebuttal

 I am satisfied with the authors' response, and increased my score to Accept. As mentioned in the discussions below, I do think it is important clarify the distinction between "fine-tuning" in the general sense versus training on domain specific data in their updated version to highlight the true setting where this study is useful.

**Claims And Evidence:**

I think most claims and the evidence presented are pretty solid. Models between 40M-1B parameter ranges are studied which is pretty comprehensive. There are extensive results on each factors and how they affect the scaling.

**Essential References Not Discussed:**

[1] https://arxiv.org/pdf/2406.01375v1 - proposes scaling laws for domain specific training after pre-training

**Experimental Designs Or Analyses:**

All experimental design choices are clear, apart from the fine tuning dataset as discussed above. The search space over the pretraining data, fine-tuning data, and model scale covers a broad range.

**Methods And Evaluation Criteria:**

- My main complaint about this paper is the notion of the fine tuning task. The authors consider sub-domain splits of the Pile, which is a dataset that is generally used as a pre-training dataset. In practice, many fine-tuning datasets and instruction tuning datasets, such as Alpaca, contain a range of examples from diverse domains.

**Other Comments Or Suggestions:**

1. I would recommend adding the hyper parameter values for each configuration and other related details for reproducibility.
2. I would also recommend adding downstream task results for these models as a sanity check. This would give a better sense of how the models perform in downstream applications before and after fine-tuning (eg. MMLU, instruction following tasks etc)

**Other Strengths And Weaknesses:**

Strengths:

Overall, a good paper that the community will find useful.

Weaknesses:

As mentioned I think the paper can be strengthened further with experiments on actual fine-tuning tasks. As such, this feels more like domain-specific training, whose scaling laws some existing work has studied [1]. I will however note that [1] has only studied domain specific pretraining i.e. in the dataset scale of billions of tokens. But, the fine-tuning setup here is still a little artificial than the type of fine-tuning that is conventionally followed (SFT/Instruction Tuning). Given this observation, I lean towards a weak accept for now.

[1] https://arxiv.org/pdf/2406.01375v1

**Questions For Authors:**

1. Could the authors elaborate on why they chose the Pile as the fine tuning task, as opposed to other SFT/Instruction-Tuning/Alignment datasets which are conventionally used?
2. From what I understand, in 4.4, the number of unique tokens available varies but they are upsampled such at $p$=1\%? If yes, it would be helpful to make this clear.

**Relation To Broader Scientific Literature:**

To the best of my knowledge, this is the first paper that studies scaling laws for fine-tuning by also considering replay data (i.e pre-training data injection) during fine-tuning. [1] studies scaling laws for forgetting during fine-tuning, but does not consider replay data. Similarly, [2] only studies scaling laws on the fine-tuning task, while this paper also attempts to model the loss on the pre-training data after fine-tuning.

[1] https://arxiv.org/pdf/2401.05605

[2] https://arxiv.org/abs/2402.17193v1

**Theoretical Claims:**

Given the empirical nature of the work, there are no explicit theory or proofs. The scaling law for fine-tuning is drawn from prior work. For forgetting equations, the choices seem fair.

---

> ### Author Rebuttal · Authors · 2025-04-01
>
> Dear reviewer,
>
> We thank you warmly for your detailed and thorough feedback on our work. We are glad to read that "The experiments are comprehensive and validate the hypotheses", that "most claims and the evidence presented are pretty solid", and that " experimental design choices are clear".  Thank you as well for the additional reference.
>
> > I would recommend adding the hyper parameter values for each configuration and other related details for reproducibility.
>
> Thanks! We added a summary table in appendix.
>
> > I would also recommend adding downstream task results for these models as a sanity check
>
> Please find [here](https://anonymous.4open.science/r/icml2025-figures-3ED2/barplot_arceasy.pdf) and [here](https://anonymous.4open.science/r/icml2025-figures-3ED2/barplot_MMLU.pdf) results for the `ARC_easy` and `MMLU` task, for the pretrained checkpoint, and for models finetuned on `dm_mathematics`. Performance degrades on the generalist questions of ARC easy, but improves on MMLU which is more aligned with `dm_mathematics`.
>
> Furthermore, on the domain `dm_mathematics` we evaluated the quality of 75 checkpoints (all model sizes above 350M) on the  `arc_easy` task, in a 0-shot setting. We take the z-score at 99% significance threshold, and report the best result below (ns = not significant, numbers are accuracy gain over p=0%). **TLDR**; the difference is significant for 15 experiments, oscillating between 3.7% and 8.7% more, and pretraining data injection always helps. As noticed before, the bigger the finetuning dataset the more important the forgetting. For these bigger datasets (>9M tokens), injecting pretraining data is crucial.
>
> |                      |    0.1% |    0.5% |      1% |      5% |
> |:---------------------|--------:|--------:|--------:|--------:|
> | ('medium', 307200)   | ns     | ns     | ns    | ns     |
> | ('medium', 921600)    | ns     | ns     | ns    | ns     |
> | ('medium', 3072000)  |   3.872 |   4.461 |   4.082 |   4.082 |
> | ('medium', 9216000)  |   4.125 |   4.082 |   4.04  | ns     |
> | ('medium', 30720000) |   4.798 |   6.566 |   4.798 |   5.429 |
> | ('large', 307200)    | ns     | ns     | ns    | ns     |
> | ('large', 921600)    | ns     | ns     | ns    | ns     |
> | ('large', 3072000)   | ns     |   3.872 |   4.588 |   5.093 |
> | ('large', 9216000)   |   3.788 |   4.966 |   5.345 |   4.588 |
> | ('large', 30720000)  |   6.944 |   7.744 |   7.239 |   8.375 |
> | ('xl', 307200)       | ns     | ns     | ns    | ns     |
> | ('xl', 921600)       | ns     | ns     | ns    | ns     |
> | ('xl', 3072000)       | ns     | ns     | ns    | ns     |
> | ('xl', 9216000)      | ns     | ns     |   3.998 | ns     |
> | ('xl', 30720000)     |   4.756 |   5.008 |   5.471 |   5.303 |
>
> > From what I understand, in 4.4, the number of unique tokens available varies but they are upsampled such at =1%? If yes, it would be helpful to make this clear.
>
> Yes you’re correct - we updated the section and [the figure](https://anonymous.4open.science/r/icml2025-figures-3ED2/ablation_pretraining_size.pdf) x-axis. Each batch is built following the process described in section 3.3: each sequence is picked from the pretraining split with probability 1%, and from the specific domain with probability 99%. The pretrained set is capped to the number of tokens wanted, and then repeated as many times as necessary.
>
> >  Could the authors elaborate on why they chose the Pile as the fine tuning task, as opposed to other SFT/Instruction-Tuning/Alignment datasets which are conventionally used?
>
>
> We wanted to simulate the setup in which a company fine-tune a model on raw data from internal documentation. The highly-specialized content of The Pile (which is partitioned semantically) allows to simulate the scenario, unlike typical typical IFT datasets that are generalist.
>
> Nonetheless, we agree that the Instruction Finetuning scenario is also of interest; thank you for your suggestion. We perform instruction finetuning on the OpenHermes dataset - 3M tokens in the train split (95% of the total). We finetune the model to perform next token prediction of the output, conditioned on the [INST] prompt input [/INST] prefix. We added [finetuning curves](https://anonymous.4open.science/r/icml2025-figures-3ED2/openhermes_finetuning.pdf) and [forgetting curves](https://anonymous.4open.science/r/icml2025-figures-3ED2/openhermes_forgetting.pdf) in appendix.
>
> We report the fitted scaling laws parameters here:
>
>
> **Finetuning**:
>
> | Domain     |   alpha |      beta |       A |      E |      HeldOutMRE |
> |:-----------|--------:|----------:|--------:|-------:|-------------:|
> | Openhermes | 0.17582 | 0.0286171 | 64.2788 | 0.4585 | 0.59% |
>
> **Forgetting**:
>
> | Domain     |    alpha |     beta |    A |    B | HeldOutMRE   |
> |:-----------|---------:|---------:|-----:|-----:|:-------------|
> | OpenHermes | 0.793364 | 0.266993 | 5513 | 8584 | 0.29\%       |
>
>
> We thank you again for your review, and hope that our answer has alleviated your concerns.

---

> > ### Comment · Reviewer_R4eh · 2025-04-01
> >
> > Thank you for the clarifications and additional experiments. I have updated my score.
> >
> > > We wanted to simulate the setup in which a company fine-tune a model on raw data from internal documentation.
> >
> > I would recommend the authors to clarify the distinction between "fine-tuning" in the general sense versus training on domain specific data in their updated version, since companies may also fine-tune on data ranging from specific domains to internal generalist instruction tuning datasets.

---

> > > ### Author Response · Authors · 2025-04-07
> > >
> > > Thank you for raising your score.
> > >
> > > > I would recommend the authors to clarify the distinction between "fine-tuning" in the general sense versus training on domain specific data in their updated version
> > >
> > > That's a good point, we will add a paragraph in introduction to emphasize that we are focusing on this setup.

---

### Official Review · Reviewer_wqv9 · 2025-03-11

**Overall Recommendation:** 2

**Summary:**

The paper studies the domain adaptation and forgetting effects of language model finetuning by deriving scaling laws that quantify these two phenomena. It shows that one can accurately predict the finetuning performance and the forgetting of the pretraining set of large language models, as a function of the model size, the number of available finetuning tokens, and of the fraction of pretraining data injected into the finetuning data mixture.

**Claims And Evidence:**

The claims made in the submission are supported by clear evidence based on my understanding of scaling law - I would suggest having people familiar with scaling law double-check the claim due to my low confidence on this area.

**Essential References Not Discussed:**

I do not have specific reference in mind.

**Experimental Designs Or Analyses:**

How the paper makes the scaling law plot is sound. However, I am wondering whether the current experiments are enough to demonstrate the robustness of the proposed scaling law:
1. The scaling law is fit with data points coming from model size of 41M, 109M, 334M, 665M, 1.27B. While the proposed curve fits these points well, the paper needs to show the scaling law can accurately predict the loss for a model that is much larger than 1.27B (due to resource constraints, the authors may consider experimenting on a 7B-parameter model).
2. While the U-curve for generalization-memorization tradeoff (Figure 2) is intuitive to interpret, I am unsure whether there is mathematical expression to characterize the U-curve. This is important for understanding the observed tradeoff can extrapolate to region not covered by the current datapoints.

**Methods And Evaluation Criteria:**

Using scaling law to understand the dynamics between the domain adaptation and forgetting effects of language model finetuning makes sense and provides novel insights.

**Other Comments Or Suggestions:**

1. Figure 7 caption: "(Zhang et al., 2024)" -> "Zhang et al., 2024"

**Other Strengths And Weaknesses:**

Strengths: The paper is novel and deepens the understanding of catastrophic forgetting in continual learning.

Weaknesses:
1. It's unsure what's the use case for the proposed scaling law. For traditional scaling laws, they could help us understand how to better distribute budget (e.g., parameter size, pre-training token size). But for scaling laws proposed in this work, the scenario is different, as the paper also shows that a small percentage of pre-training data could already significantly mitigate the forgetting.
    - I am willing to adjust my assessment if this point could be addressed.

**Questions For Authors:**

See my comment above.

**Relation To Broader Scientific Literature:**

To the best of my knowledge, this is the first paper that studies scaling law in continual learning and catastrophic forgetting.

**Theoretical Claims:**

Yes. I have carefully read the figures related to the claim that there are scaling laws that quantify the dynamics between the domain adaptation and forgetting effects of language model finetuning from various target domains.

---

> ### Author Rebuttal · Authors · 2025-04-01
>
> Dear reviewer,
>
> We thank you warmly for your detailed and thorough feedback on our work. We are glad to read that "The claims made in the submission are supported by clear evidence" and that our method "makes sense and provides novel insights."
>
> > While the proposed curve fits these points well, the paper needs to show the scaling law can accurately predict the loss for a model that is much larger than 1.27B
>
> This is indeed an important point, and we are happy to report that our scaling laws allow us to *extrapolate*, that is, as you mention, predict large scale behavior from small scale.
>
> We could predict the forgetting for **650M and 1.3B models**, on **9M and 30M unique tokens**, with a bootstrapped MRE of 0.83% - **using only models no bigger than 350M parameters, and finetuned with no more than 3M unique tokens**. We added [this table](https://anonymous.4open.science/r/icml2025-figures-3ED2/extrapolate.png) in the paper.
>
> > I am unsure whether there is mathematical expression to characterize the U-curve
>
> We agree that’s a fruitful research question. However, finding how many steps/repetitions are required to find the bottom of the U-curve is by no means trivial, since it depends on both the domain, the model size, and the number of tokens. Nonetheless, assuming fixed number of epochs, we benchmarked the scaling laws of “Scaling Data-Constrained Language Models”. We found it yielded accurate predictions, correct in our setting too. We will clarify this in the paper.
>
> Therefore, it might be possible to predict when the bottom of U-curve will be reached (find the **argmin** over time) based on the performance improvements after a few epochs. Our current work doesn’t need to characterize the full U-curve: it is sufficient to estimate the minimum value reached at the bottom (find the **min** over time).
>
> > It's unsure what's the use case for the proposed scaling law
>
> We will clarify two important outcomes from our scaling laws.
>
> A) The proposed scaling law allows to quantify forgetting as function of scale, and notably to extrapolate, as you mentioned before. Testing at small scales allows you to not be surprised with what happens at larger scales.
>
> B) We also believe our work brings some understanding to the phenomenon of forgetting. An interesting consequence of our functional form reveals that the leading cause behind forgetting might be related to the parameters count $BpN$. This suggests that forgetting is primarily due to limitations in network capacity. This is also confirmed by the fact that smaller models suffer the most: they lose up to 95% (!) of the pretraining progress when forgetting (i.e, the pretraining validation loss reverts to a point reached at 5% of pretraining), while bigger models only lose 20% of the progress. We added [this plot](https://anonymous.4open.science/r/icml2025-figures-3ED2/arxiv_forget_hours.pdf) in the paper, which quantifies forgetting with a fraction of pre-training cost lost.
>
> Overall, this balances some findings of “Beyond Chinchilla-Optimal: Accounting for Inference in Language Model Scaling Laws”, which suggested that smaller models should be preferred due to their lower inference cost. We will make these points in the paper.
>
> We thank you again for your review, and we hope that we have alleviated your concerns !

---

### Official Review · Reviewer_5seN · 2025-03-12

**Overall Recommendation:** 4

**Summary:**

This paper studies a setting (examined previously by Liu 2022, Kang et al 2024, Ibrahim et al 2024) where a small amount of pre-training data is injected in fine-tuning to prevent catastrophic forgetting of the pre-training domain and provide regularization in the target domain. In this setting, the paper develops scaling laws for both pre-training and fine-tuning losses during fine-tuning. The paper builds upon prior work (Kalajdzievski et al 2024), who develop scaling laws for forgetting losses during fine-tuning when pre-training data is not injected during fine-tuning. Results are reported using GPT-2 models of different scales on a number of datasets. The paper provides a scaling law for pre-training loss as a function of model size, number of target tokens, proportion of pre-training data added to the data mixture at fine-tuning, and also a law for the fine-tuning loss. Similar to prior work, the paper finds that a multiplicative form works better than an additive form. The paper shows that injecting even 1% of the pretraining data can prevent a degradation of the pretraining loss and can provide some regularization on the target domain.

## update after rebuttal”
The authors have addressed many of my concerns in their rebuttal especially wrt the following:
* Providing an additional isocurve
* Reporting the training loss as a function of 'Unique pretrain tokens per unique finetune token'
Hence I am increasing the score.

**Claims And Evidence:**

The claims are mostly reasonable except for the following:

* The paper does not explore the full grid of model size and data set size i.e. (N, D) but rather an isocurve D=100N. While this is a reasonable choice for computational reasons, it would have been useful to study the full grid at least for a single domain to understand how far the isocurve is from the optimal value for this domain.
* Though the paper presents a large number of results for a single model (Gpt-2), it is unclear if these laws will carry over to alternative LLMs.

**Essential References Not Discussed:**

None

**Experimental Designs Or Analyses:**

Yes, I checked. They seem reasonable.

**Methods And Evaluation Criteria:**

The methods and evaluation criteria make sense.

**Other Comments Or Suggestions:**

None

**Other Strengths And Weaknesses:**

Strengths:
* Studies a problem which is valuable to the community: how to fine-tune while preventing catastrophic forgetting.
* Develops scaling laws for both fine-tuning and pretraining loss as a function of model size, finetuning data size and proportion of pretraining data injected during fine-tuning. This is a setting that has not been examined previously in the context of developing scaling laws.
* Proposes a practical solution to prevent catastrophic forgetting and obtain some regularization by injecting a small proportion of pre-training data during fine-tuning.
* Examines the effect of pre-training data repetitions on forgetting


Weaknesses:
* Prior work (Kalajdzievski et al 2024) has developed scaling laws for forgetting for fine-tuning when using LoRA. Hence the novel aspect of this work is limited to a) obtaining these laws in a setting where a small proportion of the pre-training data is injected during fine-tuning, b) using full-finetuning rather than LoRA.
* Though the paper presents a large number of results for a single model (Gpt-2), it is unclear if these laws will carry over to alternative LLMs.
* The paper does not explore the full grid of model size and data set size i.e. (N, D) but rather an isocurve D=100N. While this is a reasonable choice for computational reasons, it would have been useful to study the full grid at least for a single domain to understand how far the isocurve is from the optimal value for this domain.
* Similar to prior work, the paper reports MRE for the additive and multiplicative scaling laws. However, it would be advantageous to discuss whether alternative functional forms of the regression model may fit the data better.
* Figure 6: It would be useful to report the extent of repetition in the pre-training dataset.

**Questions For Authors:**

None

**Relation To Broader Scientific Literature:**

* Building upon prior work (Liu 2022, Kang et al. 2024, Ibrahim et al. 2024), this paper investigates a scenario where a small proportion of pre-training data is incorporated in fine-tuning mitigate catastrophic forgetting and improve target domain regularization.
* Prior work (Kalajdzievski et al 2024) has developed scaling laws for forgetting for fine-tuning when using LoRA. Hence the novel aspect of this work is limited to a) obtaining these laws in a setting where a small proportion of the pre-training data is injected during fine-tuning, b) using full- finetuning rather than LoRA.

**Theoretical Claims:**

The paper does not make any theoretical claims.

---

> ### Author Rebuttal · Authors · 2025-04-01
>
> Dear reviewer,
>
> We thank you warmly for your detailed and thorough feedback on our work. We are glad to see that you found that "claims are mostly reasonable", that "the methods and evaluation criteria make sense," and that we study "a problem which is valuable to the community".
>
>
>  > Prior work (Kalajdzievski et al 2024) has developed scaling laws for forgetting for fine-tuning when using LoRA. Hence the novel aspect of this work is limited to a) obtaining these laws in a setting where a small proportion of the pre-training data is injected during fine-tuning, b) using full-finetuning rather than LoRA.
>
> Additionally to those differences, as explained L148, we want to point out that we use many more specialization tasks, model scales, and we measure forgetting through the pretraining loss (which is impossible to do with models like LLAMA 2 for which the training data is unavailable).
>
> > Though the paper presents a large number of results for a single model (Gpt-2), it is unclear if these laws will carry over to alternative LLMs.
>
> Indeed, we acknowledge that we only consider autoregressive decocoder-only transformers. Note that the architecture we use is widely used, up to minor changes (see e.g. sec.2.2 of [1]). Testing the impact of these minor architectural changes like use of RoPE and swiglu on the scaling laws is indeed an interesting future research direction.
>
> > The paper does not explore the full grid of model size and data set size i.e. (N, D) but rather an isocurve D=100N. While this is a reasonable choice for computational reasons, it would have been useful to study the full grid at least for a single domain to understand how far the isocurve is from the optimal value for this domain.
>
> Thanks for raising this point ! We tested our idea with another **isocurve D=10N**. We finetune on free_law,  and we measure a bootstrapped MRE of **0.57% for forgetting** and **1.14%  for finetuning**. We aded [this curve](https://anonymous.4open.science/r/icml2025-figures-3ED2/forgetting_isocurve10_freewlaw_0.5perc.pdf) and [this curve](https://anonymous.4open.science/r/icml2025-figures-3ED2/finetuning_isocurve10_freelaw_0.5perc.pdf) to the paper.
>
> > it would be advantageous to discuss whether alternative functional forms of the regression model may fit the data better
>
> We are looking for a forgetting law that (i) should yield a zero finetuning when $D^{ft}$ is zero (ii) does not exhibit too many arbitrary parameters.  For example, we found that the $+E$ term is entirely explained by the re-warming of the model (since the finetuning LR was x3 times terminal LR of pretraining). As a matter of fact, measuring it that way instead of regressing its value __decreases__ the MRE from 0.48% to 0.40%. See [this table here](https://anonymous.4open.science/r/icml2025-figures-3ED2/estimate_E.png).
>
> We also tested additive laws, as mentioned in p8. Other laws, like $A\frac{D^{\beta}(1-p)^{\kappa})}{N^{\alpha}}+E$ had a higher error of 0.67% despite having one more parameter $\kappa$. We added these results in appendix.
>
> Furthermore, to strengthen the statistical significance of our results, we also compute and report the bootstrapped MRE (we sample 125 measurements with replacement from the pool, and average the result over 128 independent samples).
>
>  > It would be useful to report the extent of repetition in the pre-training dataset
>
> Good call! Since that the bottom of the U-curve is typically reached within a few dozen epochs, and that p=1%, the number of repetitions quickly falls to 1. We updated [the figure](https://anonymous.4open.science/r/icml2025-figures-3ED2/ablation_pretraining_size.pdf) with “Unique pretrain tokens per unique finetune token” in x-axis. We discover that 0.3 unique pretrain tokens per unique finetune token are typically sufficient.
>
>
> We hope that our answers have alleviated your concerns, and we thank you again for your review !
>
> [1] Touvron, Hugo, et al. "Llama: Open and efficient foundation language models." arXiv preprint arXiv:2302.13971 (2023).

---

### Official Review · Reviewer_QR3Z · 2025-03-17

**Overall Recommendation:** 3

**Summary:**

The paper addresses two key challenges in finetuning large language models: (1) overfitting when target domain data is limited and (2) forgetting of pretraining knowledge as the model drifts from its original parameters. The paper studies pretraining data injection as a solution to these challenges, and quantifies its effects through scaling laws.

**Claims And Evidence:**

The paper's claims are supported by the evidence presented. The central claim that 1% pretraining data injection mitigates forgetting is clearly demonstrated through experiments across multiple model sizes and domains.

Scaling law predictions match observed values fairly closely.

**Essential References Not Discussed:**

N/A

**Experimental Designs Or Analyses:**

The experimental setup is sound.

**Methods And Evaluation Criteria:**

The methods are appropriate for the research question. The choice to measure forgetting via pretraining loss is reasonable. The experimental design systematically varies model size, finetuning dataset size, and pretraining injection proportion.

**Other Comments Or Suggestions:**

- The use of the word "fine-tuning" throughout the paper may be slightly misleading as it's more commonly associated with post-training (SFT, RLHF...). The paper would benefit from clearer positioning relative to instruction tuning.

**Other Strengths And Weaknesses:**

- The paper addresses a practical problem in language model finetuning with a simple and effective solution (1% pretraining data injection).
- The experimental design is systematic and thorough, covering multiple model sizes, domains, and dataset sizes.
- The finding that smaller models are more prone to forgetting than larger ones is interesting.
- The paper focuses on continual pre-training scenarios rather than what is commonly called "fine-tuning" in modern LLM contexts.

**Questions For Authors:**

- How sensitive is the 1% rule to the nature of the pretraining data? Would you expect higher or lower optimal injection rates if the pretraining data is highly diverse but the target domain is specialized?
- How sensitive is the 1% rule to data quality? For example, would it be possible to get away with 0.1% of the pretraining data if it was carefully selected? This question falls outside the scope of this paper, but it could be an interesting direction to take this line of inquiry.
- Did you observe any qualitative differences in the types of knowledge that were forgotten when p=0% versus preserved when p=1%? Are certain types of knowledge (e.g., factual, linguistic, reasoning) more prone to forgetting?
- Your scaling law includes parameter B, which indicates the relative efficiency of parameters allocated to pretraining versus finetuning. The values vary significantly across domains. What factors do you believe drive these significant differences?
- Your results suggest that pretraining data injection is a regularizer that improves generalization. How does it compare to standard regularization methods (e.g., L2 regularization) at preventing forgetting?
- Is there a way you can test some of these ideas on standard Llama/Gemma/Qwen base checkpoints?

**Relation To Broader Scientific Literature:**

The paper's contributions build on work in scaling laws and compute-optimal training. It is also closely related to the literature on catastrophic forgetting and data mixtures. It has a central point that is, to my knowledge, new and relevant for continual LLM pretraining.

**Theoretical Claims:**

N/A

---

> ### Author Rebuttal · Authors · 2025-04-01
>
> Dear reviewer,
>
> We thank you warmly for your detailed and thorough feedback on our work. We are happy to read that you found that our work is a “simple and effective solution”, that "claims are supported by the evidence presented", and that we have "a central point that is [...] new and relevant for continual LLM pretraining".
>
> > The paper focuses on continual pre-training scenarios rather than what is commonly called "fine-tuning" in modern LLM contexts.
>
> We will clarify this very important point in the paper. A critical part of our setup is the specialization data scarcity: we only have a small number of tokens to train on, which is not necessarily the case in continual pre-training, while it is a defining aspect of fine-tuning.
> **Our findings also extend to more standard LLM fine-tuning set like openHermes**, please refere to the last part of the answer to rev.R4eh
>
> > How sensitive is the 1% rule to the nature of the pretraining data?
>
> This is an interesting question; testing it would require using a novel pretraining set and pretraining models on it, which is cumbersome. However, we believe that indeed the diversity of the pretraining data is very important ot counteract forgetting.
>
> > How sensitive is the 1% rule to data quality?
>
> Pretraining data injection acts as a regularization. The value of $p$ allows to move along a Pareto front - see [the figure](https://anonymous.4open.science/r/icml2025-figures-3ED2/pareto_front.pdf). We expect that data quality does not have much impact on the fraction p necessary to overcome forgetting, since p only drives the "regularization strength". However, we believe that higher data quality in the pretraining set would allows using less of it to fine tune in a scenario where we repeat the pretraining data; we expect improvements in an experiment like fig.6
>
> > Did you observe any qualitative differences in the types of knowledge that were forgotten? Are certain types of knowledge more prone to forgetting?
>
> We did not look into the details of knowledge that are forgotten; but clarifying how models forget is an extremely interesting future research direction.
>
> > Your scaling law includes parameter B [...]. The values vary significantly across domains. What factors do you believe drive these significant differences?
>
> This is an interesting point. The parameter B indicates how much pretraining data helps mitigating forgetting. A high B value therefore indicates a strong discrepancy between the pretraining and fine-tuning set, ann indeed B is low for datasets such as free law and wikipedia which are close to the pretraining set, while it is high for dm_mathematics and euro parl, which are far from the pretraining set. We will clarify this in the text.
>
> > Your results suggest that pretraining data injection is a regularizer that improves generalization. How does it compare to standard regularization methods (e.g., L2 regularization) at preventing forgetting?
>
> We tested an "anchored" baseline with a variant of AdamW, using $\lambda(\theta\_t-\theta\_0)$ as the weight decay term, instead of the conventional $\lambda\theta_t$, where $\theta\_0$ are the parameters of the pretrained checkpoint. Note that this is equivalent to using weight decay on the delta between fine-tuned and base model. We added [this illustration](https://anonymous.4open.science/r/icml2025-figures-3ED2/demo_anchored_adamw.pdf) to the paper. For $\lambda\in[1e-2, 1e-1, 1]$ the forgetting remains significant (between 15% and 4% more than p=1%, accross all model sizes). For $\lambda\geq 1$ the finetuning performance decreases compared to the baseline p=1%. Therefore _more data diversity is superior to regularization in the parameter space_. We added [the plot](https://anonymous.4open.science/r/icml2025-figures-3ED2/github_anchored_adamw.pdf) in appendix.
>
> We also found that standard weight decay had not impact on forgetting.
>
> > Is there a way you can test some of these ideas on standard Llama/Gemma/Qwen base checkpoints
>
> Thank you for the suggestion, this is an interesting future work. We chose to focus on a family of models we had full control over, where we precisely knew the pretraining pipeline and datasets. This allowed us to observe clear trends as a function of scale. In contrast, the precise training pipeline as well as their training data are not available.
>
> However, we tested our ideas with other models from the same family, with the **isocurve D=10N**. We finetune on free_law,  and we measure a bootstrapped MRE of **0.57% for forgetting** and **1.14%  for finetuning**. We aded [this curve](https://anonymous.4open.science/r/icml2025-figures-3ED2/forgetting_isocurve10_freewlaw_0.5perc.pdf) and [this curve](https://anonymous.4open.science/r/icml2025-figures-3ED2/finetuning_isocurve10_freelaw_0.5perc.pdf) to the paper.
>
> We hope that our answers have alleviated your concerns, and we thank you again for your review !

---

### Decision · Program_Chairs · 2025-05-01

**Decision:**

Accept (poster)

**Comment:**

The paper derives "scaling laws" on how a finetuned model either overfits to the target task, or "drifts away" from the pre-trained model. One natural idea to "fix" such an issue is to add more of pre-training data; the paper shows that adding a small amount of such data helps prevent drifting and overfitting. The reviewers are generally positive about the evaluation and methodology. The problem itself is an interesting one, of broad appeal.